# The origin and adaptive evolution of domesticated populations of yeast from Far East Asia

Shou-Fu Duan [1,2], Pei-Jie Han[1], Qi-Ming Wang[1], Wan-Qiu Liu[1], Jun-Yan Shi[1,2], Kuan Li[1], Xiao-Ling Zhang[1] & Feng-Yan Bai[1,2]

The yeast *Saccharomyces cerevisiae* has been an essential component of human civilization because of its long global history of use in food and beverage fermentation. However, the diversity and evolutionary history of the domesticated populations of the yeast remain elusive. We show here that China/Far East Asia is likely the center of origin of the domesticated populations of the species. The domesticated populations form two major groups associated with solid- and liquid-state fermentation and appear to have originated from heterozygous ancestors, which were likely formed by outcrossing between diverse wild isolates primitively for adaptation to maltose-rich niches. We found consistent gene expansion and contraction in the whole domesticated population, as well as lineage-specific genome variations leading to adaptation to different environments. We show a nearly panoramic view of the diversity and life history of *S. cerevisiae* and provide new insights into the origin and evolution of the species.

[1] State Key Laboratory of Mycology, Institute of Microbiology, Chinese Academy of Sciences, Beijing 100101, China. [2] College of Life Sciences, University of Chinese Academy of Sciences, No.19(A) Yuquan Road, Shijingshan District, Beijing 100049, China. Correspondence and requests for materials should be addressed to F.-Y.B. (email: baify@im.ac.cn)

The budding yeast *Saccharomyces cerevisiae* has been used worldwide for baking, brewing, distilling, and winemaking for tens of centuries. The earliest evidence for fermented wine-like beverage production dates back to Neolithic times about 9000 years ago in China[1]. *S. cerevisiae* has also been extensively studied as a model in physiology, genetics, and molecular biology and became the first eukaryote to have its genome completely sequenced[2]. However, until recently, substantially less effort has centered on the domestication of yeast, in comparison with the long history and extensive study on the domestication of plants and animals since Darwin[3,4]. The lag is partially due to the fact that early research on *S. cerevisiae* focused only on a few laboratory strains and that we know very little about the ecology and natural history of the yeast[5].

A decade ago, the first molecular study supporting the phylogenetic separation between wild and domesticated populations of yeast was reported[6], and since then a variety of phylogenetically distinct lineages of *S. cerevisiae* from nature and human-associated environments have been recognized[7,8]. However, there have been different opinions on the fundamental question of whether the diversity of *S. cerevisiae* is primarily shaped by niche adaptation and selection or neutral genetic drift[9]. Many studies support selection and ecology being the primary driver for the evolution of *S. cerevisiae*[10–19]; however, other studies emphasized the role of neutral genetic drift[9,20–23]. The origin of domesticated lineages of *S. cerevisiae* also remains to be resolved. Recent studies have recognized major domestication events in the history of *S. cerevisiae* for beer, sake, and wine fermentation[6,8,18–20,24–26]. Different hypotheses for the origin of wine strains from Africa[6], Mesopotamia[24], and Mediterranean[27] have been proposed. A recent population genomics study focusing on ale beer and wine yeasts revealed that today's industrial yeasts originated from only a few ancestors[18]. However, the center of origin and the closest wild relative of the domesticated population of *S. cerevisiae* are still uncertain.

The uncertainty or inconsistency about the evolutionary history of *S. cerevisiae* revealed in previous studies could be due to insufficient or biased sampling. The *S. cerevisiae* strains compared in previous studies are mostly domesticated ones from man-made environments. A recent large-scale field survey clearly showed that *S. cerevisiae* occurs in highly diversified substrates from man-made environments to other habitats remote from human activity, such as primeval forests[28]. Primeval forests within China harbor highly diverged wild lineages of *S. cerevisiae*, including the oldest lineages of the species documented so far. The Chinese wild lineages exhibited nearly double the combined genetic variation identified in *S. cerevisiae* strains sampled from the rest of the world. This result combined with other studies[29–32] suggests that China, or more broadly Far East Asia, is likely the origin center of *Saccharomyces* yeasts. Therefore, the wild and domesticated populations of *S. cerevisiae* from this area are indispensable for illuminating the evolutionary history of the species.

Here, we analyzed a set of 106 wild and 160 fermentation-associated *S. cerevisiae* isolates from diversified sources in China (Supplementary Data 1), including the oldest wild lineages from primeval forests and unwittingly domesticated lineages involved in ancient fermentation processes. This collection of isolates represents the largest genetic diversity of *S. cerevisiae* documented so far. We performed high coverage genome resequencing and phenotypic characterization of these isolates in their natural ploidy (Supplementary Data 2) and carried out an integrated phylogenomic analysis by incorporating worldwide *S. cerevisiae* isolates sequenced in previous studies[17,18,20,27]. We find that China/Far East Asia is likely the center of origin of the domesticated populations of *S. cerevisiae*. The domesticated populations, which are exclusively heterozygous appear to have originated from ancestors formed by outcrossing between diverse wild isolates primitively for adaptation to maltose-rich niches, and then have undergone extensive genome evolution through gene expansion, contraction, introgression, and horizontal gene transfer, leading to adaptation to specific fermentation environments.

## Results

**The domesticated yeast isolates belong to two major groups.** A phylogenetic tree was constructed first for the 266 *S. cerevisiae* isolates sequenced in this study based on the maximum likelihood analysis of the high-quality whole genome single nucleotide polymorphisms (SNPs), covering a total of 923,479 sites. The tree shows that the wild and fermentation-associated isolates are clearly separated (Fig. 1a). The wild isolates were clustered into ten clear lineages, largely recapitulating the result of our previous multilocus phylogenetic analysis[28]. All the isolates from primeval forests were included in six basal wild lineages CHN-I, CHN-II, CHN-III, CHN-V[28], and two new lineages CHN-IX and CHN-X. CHN-IX contains isolates from a subtropical primeval forest located in central China and represents the most basal lineage of *S. cerevisiae* as resolved by using *Saccharomyces paradoxus* as the out group. The isolates from secondary forests, orchards and fruit clustered mainly into four lineages CHN-IV, CHN-VI/VII, CHN-VIII, and Wine. The latter includes four Chinese isolates from grape and orchards (Supplementary Data 1), which cluster together with European wine strains[20,28].

The isolates from fermentation environments (Supplementary Note 1) were separated into two major groups. The isolates associated with solid-state fermentation processes, including Mantou (steamed bread), Baijiu (Chinese distilled liquors), Huangjiu (rice wines), and Qingkejiu (highland barley wines), were all located in one major group with 100% bootstrap support. Ten lineages were recognized from this group. Isolates associated with Baijiu, Huangjiu, and Qingkejiu fermentation formed three distinct lineages. Those associated with Mantou fermentation were located into seven different lineages, which are designated as Mantou 1–7 (Fig. 1a, Supplementary Data 1). The isolates associated with milk and molasses fermentation formed two separate lineages Milk and ADY (active dry yeast), respectively. These two lineages were located in the other major group together with the Wine lineage. This group is apparently associated mainly with liquid-state fermentation (Fig. 1a).

The main topologies of the trees and the clustering of the wild and domesticated Chinese isolates remained stable when an additional 287 isolates with worldwide origins[17,18,20,27] were added. The separation of the wild from the domesticated populations and the solid from the liquid-state fermentation groups was again clearly observed (Supplementary Figs. 1 and 2). Japanese Sake and Chinese Huangjiu are similarly produced by solid or semisolid-state fermentation of rice and, interestingly, the Sake strains were all clustered in the Huangjiu lineage. The majority of the wine strains clustered in a single lineage together with the four Chinese strains from fruit and orchards in the liquid fermentation group. The two lineages of ale beer isolates (Beer 1 and Beer 2) and the Mixed lineage[18] were resolved as separate lineages in the liquid-state fermentation group in this study (Supplementary Fig. 1). The Beer 1 and the Mixed lineages formed a subgroup together with the ADY lineage, while the Beer 2 together with the Milk and the Wine lineages formed another subgroup (Supplementary Figs. 1 and 2). The close relationship of the Mediterranean oak (MO) lineage with the Wine lineage as shown in a previous study[27] was also resolved here (Supplementary Fig. 2).

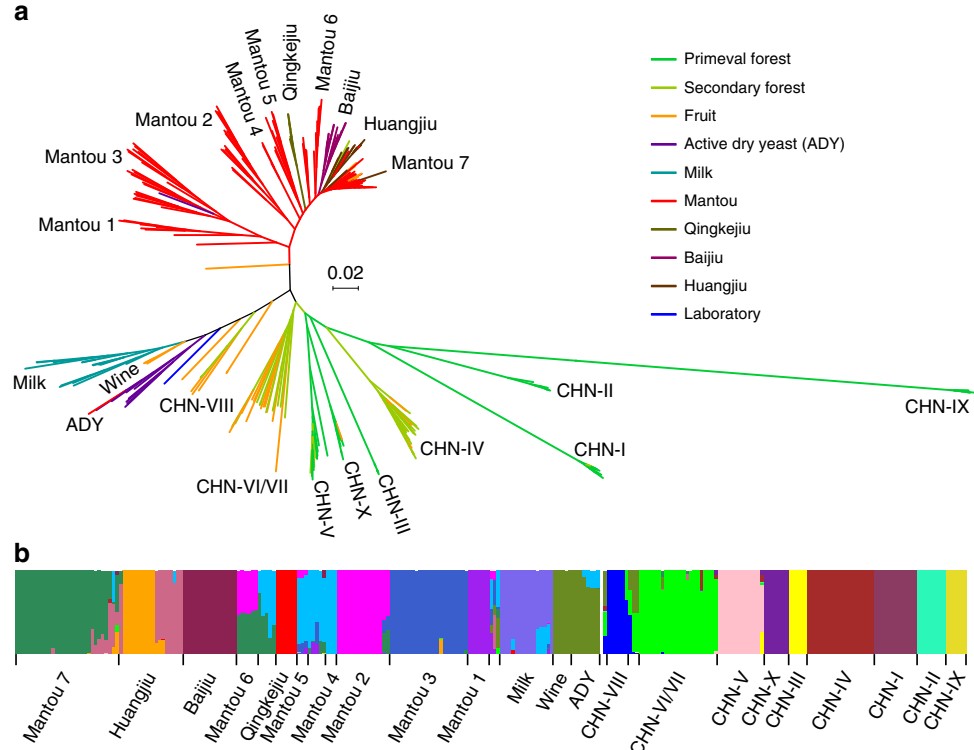

**Fig. 1** Phylogeny and population structure of wild and domesticated isolates of *S. cerevisiae* from China. **a** Unrooted tree from maximum likelihood analysis based on 923,479 genome-wide SNPs. Bootstrap support values to each lineage and major clade are 100% except for the Mangtou 7 lineage, which is supported by 80% bootstrap resampling. Scale bar, 0.02 substitutions per site. Strain branches are colored according to ecological origins. **b** Population structure inferred using the ADMIXTURE program based on 895,428 biallelic SNPs when *K* (the number of populations assumed) is set to 20 as determined by the minimum cross-validation error check. Each strain is represented by a single vertical line broken into 20 coloured segments, with lengths proportional to each of the 20 inferred clusters. Isolates in each identified population are consistent with the lineages identified in the phylogenetic tree presented in **a**

Wild isolates from other Far East Asian countries are usually located in basal wild lineages; while those from other regions of the world usually clustered in wild lineages with a closer relationship to domesticated lineages (Supplementary Fig. 1). The Malaysian strains from rainforests formed a distinct lineage closely related with lineage CHN-X and an oak strain (YJM1418) from Japan was located in lineage CHN-IV. The West African strains were located on branches basal to lineage CHN-VIII and the North American oak strains were clustered in lineage CHN-VI/VII (Supplementary Fig. 1).

In the population structure analysis[33] of the 266 isolates sequenced in this study, a maximum resolution was achieved when *K* was set to 20 (Fig. 1b and Supplementary Fig. 3a, c); while for the expanded dataset including additional 287 isolates sequenced in previous studies[17,18,20,27], a maximum resolution was achieved when *K* was set to 27 (Supplementary Figs. 1 and 3b, d). All the wild lineages and the majority of the domesticated lineages were resolved to be distinct populations though different degrees of recombination were observed in the wild lineages from secondary forests and fruit and in the domesticated lineages. Notably, lineage Mantou 6 shares polymorphisms with lineages Mantou 2 and Mantou 7 and lineage Huangjiu contains two subclusters (Fig. 1 and Supplementary Fig. 1).

The sequence diversity of the wild isolates ($\pi = 8.08e-3$) is significantly higher than (nearly double) that of the domesticated isolates ($\pi = 4.22e-3$, $P < 0.0001$) in China, as calculated from genome wide SNPs (Supplementary Data 3), though the geographic distribution of the domesticated isolates is apparently wider than that of the wild isolates (Supplementary Fig. 4). The

maximum inter-lineage sequence divergence (1.64%) of *S. cerevisiae* was found between lineages CHN-IX and Milk. A principle component analysis (PCA) also showed that the wild lineages were clearly separated from each other, while the domesticated lineages were usually clustered together (Supplementary Fig. 5). The shared polymorphisms between the two major domesticated groups (33.5%) and between different domesticated lineages (11.7 % on average) were significantly higher than those between different wild lineages (1.8% on average) ($P < 0.001$) (Supplementary Data 4).

The results obtained imply the existence of a bottleneck in the evolutionary history of the domesticated population from the wild population of *S. cerevisiae* in China. We then performed a demographic analysis based on non-coding SNPs from isolates representing all the wild and domesticated lineages recognized in this study (Supplementary Fig. 6). The result showed that the fractions of the estimated effective size of the ancestral population that entered into the wild and the domesticated groups were 99.36% and 0.64%, respectively. The migration from the wild to the domesticated group (0.525) was significantly higher than (seven times) that (0.072) of the migration from the domesticated to the wild group (Supplementary Fig. 6). These data support a bottleneck event during the domestication history of yeast in China.

**Pronounced difference in heterozygosity and sexuality.** A striking difference in heterozygosity was observed between the wild and domesticated isolates (Fig. 2, Supplementary Fig. 7a and Supplementary Data 2). The wild isolates from primeval forests as

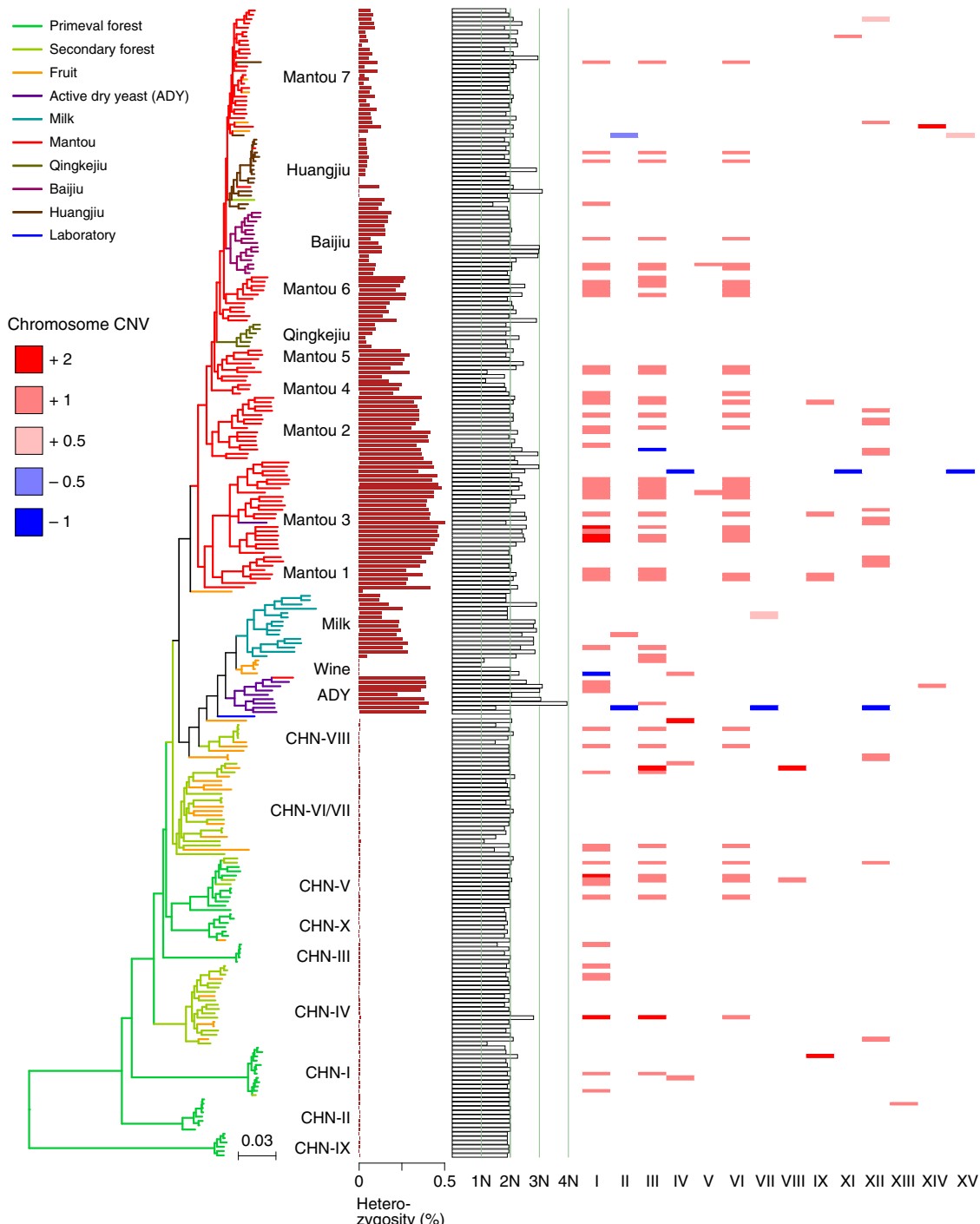

**Fig. 2** Heterozygosity, ploidy, and chromosome copy-number variation (CNV) in wild and domesticated isolates of *S. cerevisiae*. Isolates are represented by terminal branches in the phylogenetic tree constructed from the maximum likelihood analysis on genome wide SNPs and are colored according to their ecological origins. Heterozygosity is expressed as the ratio of heterozygous SNPs to the consensus genome size of each isolate. Ploidy of each isolate was estimated using flow cytometry and is represented by a bar graph. Amplification of individual chromosomes above 2n is shown by a pink or red stripe with a digital value (pink 0.5, amplification of a part of the chromosome; light red 1, one extra copy of the whole chromosome; dark red 2, two extra copies of the whole chromosome). Deletion of individual chromosome below 2n is shown by a blue stripe with a digital value (light blue −0.5, deletion of a part of the chromosome; dark blue −1, deletion of one copy of the whole chromosome)

well as from secondary forests, orchards and fruit are almost homozygous, with an average ratio of heterozygous sites of 0.0055% in the genome (ranging from 0.0023 to 0.0147%). Almost all the domesticated isolates from fermentation environments are heterozygous, with an average ratio of heterozygous sites of 0.2144% (ranging from 0.0024 to 0.5080%, $P = 1.1e-38$).

The sporulation rates of the wild and domesticated populations also differ significantly (Supplementary Fig. 7b). The majority of the wild isolates sporulate well, with an average sporulation rate of 60%. In contrast, most of the domesticated isolates were unable to sporulate, showing an average sporulation rate of 14% ($P = 1.3e-23$). Furthermore, the sporulated domesticated isolates

usually exhibited very low spore viability. The average spore viability rates of the wild isolates tested were 95.2%, while that of the domesticated isolates tested were 18.8% ($P = 1.8e-05$) (Supplementary Fig. 7c). The sporulation and spore viability rates are negatively correlated with heterozygosity, with a Spearman rank correlation of $-0.73$, consistent with Magwene et al.[34].

**Aneuploidy is common in wild and domesticated populations.** The ploidies of the 266 Chinese wild and domesticated isolates were determined based on the combination of flow cytometry (determining the relative DNA content per cell) and sequence coverage (copy-number variation (CNV)) data as shown in Supplementary Fig. 8. The ploidies of the isolates tested vary from haploidy to tetraploidy (Fig. 2 and Supplementary Data 5). As expected, the majority (233, 87.6%) of the isolates have a basal diploid genome and only 12 (4.5%), 20 (7.5%), and one (0.4%) isolate has a basal haploid, triploid and tetraploid genome, respectively. A total of 181 (68.0%) isolates are euploid and the remaining 85 (32.0%) isolates are aneuploid. The aneuploid isolates occur in similar frequency in wild (28/106 = 34.0%) and domesticated (57/160 = 35.6%, $P = 0.115$) isolates. A total of 30 aneuploidy patterns with various copy number variations of different chromosomes were identified from the aneuploid isolates.

Chromosome duplication is much more common than chromosome deletion in the aneuploid isolates. Chromosome deletion was observed in only four (4.7%) of the 85 aneuploid isolates. Notably, among the aneuploid isolates identified, 54 (63.5%), 43 (50.6%), and 36 (42.4%) isolates have one to two extra copies of the smallest chromosomes I, III, and VI, respectively; and extra copies of these three chromosomes occur simultaneously in 33 (38.8%) of the aneuploid isolates (Fig. 2 and Supplementary Data 5).

**Gene expansion and contraction in domesticated populations.** We detected a total of 225 genes that showed significant copy number variation ($P < 0.01$) between wild and domesticated groups or between individual lineages, including 105 genes with known functions and 120 genes with unknown functions (Fig. 3, Supplementary Fig. 9, and Supplementary Data 6). The patterns of expansion or contraction of these genes are largely associated with domestication and adaptation to specific niches. The genes with known functions showing significant CNV are usually associated with environmental stress response; sugar transportation and metabolisms; and amino acid transportation (Supplementary Data 6, Supplementary Note 2).

A considerable number of genes associated with stress response, including the *ARR* gene cluster involving resistance to

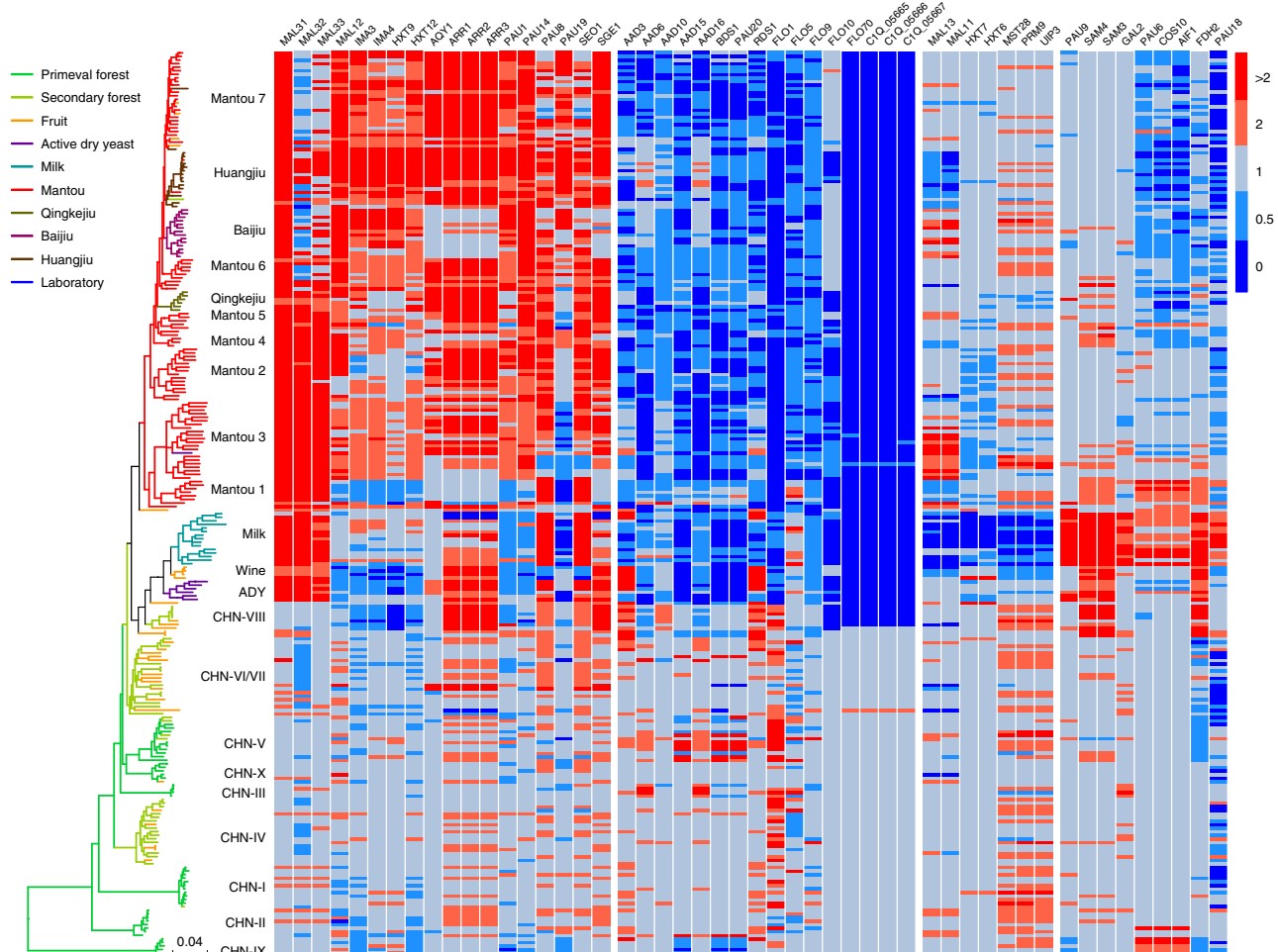

**Fig. 3** Copy-number variation (CNV) of selected genes in wild and domesticated isolates of *S. cerevisiae*. Gene names are given at the top of the heat map. Isolates are represented by terminal branches in the phylogenetic tree constructed from the maximum likelihood analysis on genome wide SNPs and are colored according to their ecological origins. Heat map colors reflect different degrees of gene duplication (red shades) or deletion (blue shades) from the basal level (grey) according to the scale on the right with strain S288c as the reference. The exact relative values of CNV are given in Supplementary Data 6

arsenic compounds[35,36], show a clear trend of expansion in the domesticated population (Fig. 3 and Supplementary Data 6). On the other hand, a considerable number of other genes associated with stress response, including *FLO* genes *FLO1*, *FLO5*, *FLO9*, *FLO10*, and *FLO70*, showed a general trend of contraction in domesticated lineages. Strikingly, *FLO70* together with three other hypothetical protein genes (Fig. 3 and nos. 161–164 in Supplementary Data 6 and Supplementary Fig. 9), which were identified from a bioethanol strain[37] are present in all wild isolates except lineage CHN-VIII but deleted in all domesticated isolates.

A considerable number of genes associated with sugar transportation and metabolism are expanded in the majority of domesticated lineages (Fig. 3 and Supplementary Data 6). Notably, genes involved in maltose utilization, including maltose transporter gene *MAL31*, maltase genes *MAL12* and *MAL32*, and a *MALx1* and *MALx2* transcription activator gene *MAL33* are duplicated in the majority of domesticated isolates. Most remarkably, *MAL31* is amplified one to two fold in almost all domesticated isolates, as compared with wild isolates.

Many genes are expanded or contracted in only specific lineages; most remarkably, in the Milk lineage (Fig. 3 and Supplementary Data 6). Genes duplicated only in the Milk lineage

include *GAL2* encoding galactose transporter (also able to transport glucose) and a few other genes (e.g., *FDH2*, *SAM3*, *SAM4*, and the *YRF1* family) with known function. More than 20 genes of unknown functions were found to be almost exclusively expanded in the Milk and ADY lineages but deleted in most wild and other domesticated isolates.

**HGT and introgression events tend to be lineage specific.** We identified a total of 79 fragments of more than 1.5 kb in length, which were regarded as representing horizontal gene transfer (HGT) or introgression events. Notably, the majority of the HGT and introgression fragments are lineage specific (Fig. 4 and Supplementary Data 7). According to the source estimation based on sequence identity (Supplementary Data 7) and phylogenetic analysis (Supplementary Fig. 10), these fragments can be classified as: (1) horizontally transferred from distantly related yeast genera; (2) introgressed from other species within the genus *Saccharomyces*; (3) introgressed from sources that likely represent a yet-to-be-discovered basal species or sibling genus of *Saccharomyces*; and (4) from totally unknown sources.

Among the HGT fragments, two (fragments 1 and 2) are from a species closely related with *Kluyveromyces lactis* (Supplementary

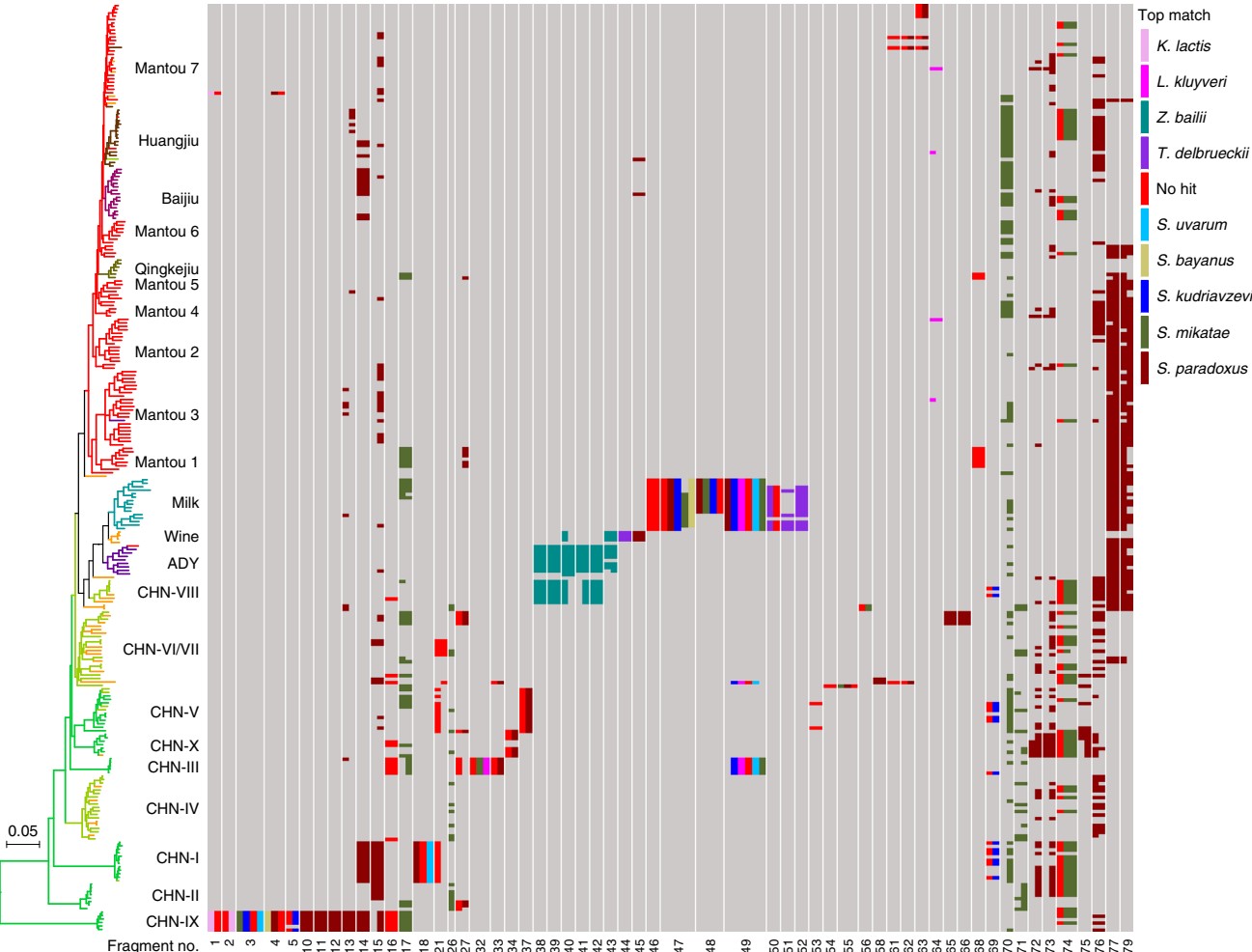

**Fig. 4** Selected introgression and horizontal gene transfer (HGT) events in wild and domesticated isolates of *S. cerevisiae*. Isolates are represented by terminal branches in the phylogenetic tree constructed from the maximum likelihood analysis on genome wide SNPs and are colored according to their ecological origin as shown in Fig. 1. The fragments (numbered at the bottom) with top matches from species other than *S. cerevisiae* are shown, which are selected from the 79 possible introgression and HGT fragments (Supplementary Data 7) detected in the 266 isolates of *S. cerevisiae* included in the tree. Top matches or putative sources of the alien fragments are shown on the upper right based on sequence identities given in Supplementary Data 7

Fig. 10a) and almost exclusively found in lineage CHN-IX; four (fragments 38–41) are apparently from *Zygosaccharomyces bailii* (Supplementary Fig. 10b), and mainly distribute in lineages CHN-VIII and ADY (Fig. 4 and Supplementary Data 7); four are most likely from *Torulaspora delbrueckii* and occur either only in the three orchard isolates of the Wine lineage (fragment 44) (Supplementary Fig. 10c), or only in isolates of the Milk lineage (fragments 50–52) (Supplementary Fig. 10d). Two of the three HGT fragments (regions A–C) first found in the wine yeast strain EC1118[10] were identified in this study. Region B (fragment 42) from *Zygosaccharomyces rouxii* exists in lineage CHN-VIII and parts of the region occur in lineage ADY. A major part (~56 kb) of region C (fragment 43) is found in lineage ADY and a minor part (~12.5 kb) of this fragment occurs in three Chinese orchard isolates in lineage Wine (Fig. 4 and Supplementary Data 7). Region A was not detected from the isolates sequenced in this study.

The introgressed fragments from other species within the genus *Saccharomyces* are also usually distributed only in single or limited lineages. *S. paradoxus*, *Saccharomyces mikatae*, *Saccharomyces bayanus*, *Saccharomyces kudriavzevii*, *Saccharomyces uvarum*, and unknown species or lineages within *Saccharomyces* were identified as possible sources with the former being the dominant donor (Fig. 4, Supplementary Fig. 10e–l, Supplementary Data 7, and Supplementary Note 3).

Phylogenetic analyses showed that some of the fragments were from unknown sources with a phylogenetic position just outside the genus *Saccharomyces*. The gene *ECM4* harbored in fragment 3 (5.5 kb in length found only in lineage CHN-IX) (Fig. 4 and Supplementary Data 7) was located in a branch basal to the known *Saccharomyces* species (Supplementary Fig. 10m). Fragment 49 (9.5 kb in length) found only in lineages CHN-III and Milk and in one isolate (YN3) of lineage CHN-VI/VII harbors the *GAL7-GAL10-GAL1* gene cluster of the galactose metabolism network. The phylogenetic tree of this gene cluster showed that the *S. cerevisiae* isolates harboring this introgressed fragment were located just outside the genus *Saccharomyces* (Supplementary Fig. 10n). These data imply the existence of a missing or yet-to-be-discovered species or genus basal to the known *Saccharomyces* species.

**Phenotypes associated with ecology and genomic variations**. We observed remarkable phenotypic variations among the wild and domesticated isolates compared, which were either generally associated with the whole domesticated population or with specific or limited lineages (Supplementary Data 8). The majority of these phenotypic variations are also clearly correlated with specific genomic variations. A clear and remarkable difference between the wild and domesticated populations is their maltose utilization ability (Fig. 5). The average growth rate (7.22e−3 OD/min) and efficiency (4.00 OD) of the domesticated isolates were significantly higher than the average growth rate (1.94e−3 OD/min, $P = 3.16e-38$) and efficiency (0.89 OD, $P = 2.44e-30$) of the wild isolates, respectively, when maltose was supplied as the sole carbon source. Almost all the domesticated isolates tested (159/160, 99.38%) can rapidly and efficiently utilize maltose, while only 17 (16.04%) of the wild isolates could utilize this sugar with a similar efficiency (4.04 OD, $P = 0.756$), but a slightly lower average rate (6.28e−3 OD/min, $P = 0.00152$) compared with the domesticated isolates (Fig. 5). These wild isolates with elevated maltose utilization ability are mostly from fruit and secondary forest and concentrated in lineage CHN-VIII and branches basal to the liquid-state fermentation group (Fig. 5). Unexpectedly, all the domesticated isolates in the Milk lineage (except isolate F3-4) also showed high maltose utilization ability, although maltose is absent in milk. The elevated maltose utilization ability of the

domesticated isolates and nine of the maltose positive wild isolates is clearly correlated with the expansion of *MAL* genes, especially *MAL31* (Fig. 5 and Supplementary Data 6).

Both the wild and domesticated isolates usually grew well in galactose (Fig. 5). Notably, the 15 isolates in the Milk lineage, the five isolates in lineage CHN-III and one isolate YN3 from fruit in lineage CHN-VI/VII showed exceptionally higher galactose utilization rates. The majority of these isolates even grew faster in galactose than in glucose, implying a shift from glucose to galactose as the most favorite sugar in these isolates (Fig. 5 and Supplementary Data 8). The isolates with elevated galactose utilization rate usually have duplicated *GAL2* genes and all possess the introgressed *GAL7-GAL10-GAL1* gene cluster (Supplementary Fig. 10n). The association of melibiose, r-affinose and sucrose utilization with specific genome or gene changes and the association of tolerance to high temperatures (40 and 41 °C) and 9% ethanol with specific lineages or environments were also observed (Fig. 5, Supplementary Data 8, and Supplementary Note 4).

## Discussion

Based on a limited sample of isolates and nuclear DNA markers, we previously showed evidence for a Far East Asian origin of *S. cerevisiae*[28, 31]. Here, we provide stronger evidence supporting this hypothesis. We found two new basal wild lineages (CHN-IX and CHN-X) from primeval forests in China with one of them (CHN-IX) being the oldest one. The discovery of lineage CHN-IX resulted in nearly one-third increase in the global genetic diversity of *S. cerevisiae* (Supplementary Data 3). Wild isolates belonging to ancient basal lineages have also been found from forests in other Far East Asian countries (Supplementary Fig. 1), but have not been found from other areas, despite extensive survey in Europe[27,38], North America[7,26], South America[39], New Zealand[21,32,40], and Africa[8,41].

We show that the genetic diversity of the domesticated population of *S. cerevisiae* in China is also much higher than that observed in other regions of the world. Previous studies have recognized only five main lineages that contain the majority of worldwide domesticated or industrial isolates of *S. cerevisiae*, namely Wine, Sake, Beer 1, Beer 2, and a Mixed lineage containing bread isolates[6,18–20,26]. Here, we identified 12 lineages from Chinese domesticated isolates: ADY, Baijiu, Huangjiu, Qingkejiu, Milk, and Mantou 1–7. The Sake lineage recognized before actually represents a subcluster in the Huangjiu lineage associated with rice fermentation. Isolates belonging to the Wine lineage are also present in fruit and orchards in China. The result suggests that China or Far East Asia is also the center of origin of domesticated populations of *S. cerevisiae*.

The domesticated lineages documented worldwide so far belong to two major monophyletic groups associated with solid- and liquid-state fermentation, respectively. Our phylogenomic analyses show that the two major domesticated groups share a common origin that diverged from the wild lineage CHN-VI/VII containing isolates from fruit, orchards and secondary forests in China. The following additional observations support this single origin hypothesis. First, there was a bottleneck as the domesticated populations diverged from the wild populations of *S. cerevisiae* (Supplementary Figs. 5 and 6). Second, almost all the domesticated isolates are heterozygous and all the wild isolates are homozygous (Fig. 2). Third, the domesticated lineages share common expansion and contraction patterns of certain genes regardless of their sources (Fig. 3, Supplementary Fig. 10, and Supplementary Data 6). The CNV patterns of maltose metabolism genes also support the single origin hypothesis. Although the isolates in the Milk and Wine lineages are from niches without

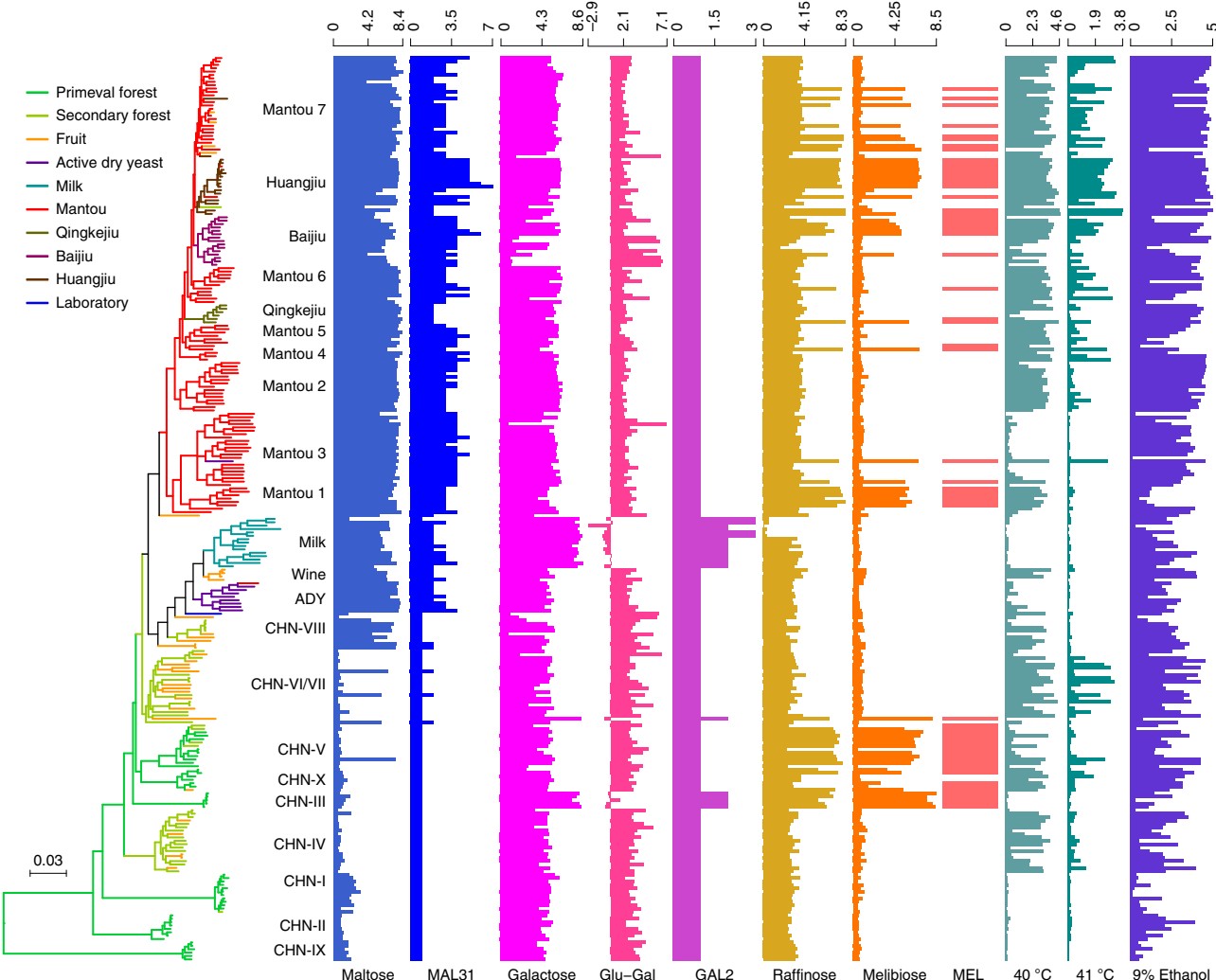

**Fig. 5** Phenotypic traits and associated gene copy-number variations (CNVs) of wild and domesticated isolates of *S. cerevisiae*. Isolates are represented by terminal branches in the phylogenetic tree constructed from maximum likelihood analysis on genome wide SNPs and are colored according to their ecological origins. Bar charts with labels at the bottom reflect phenotypic traits or CNVs. Maltose, Galactose, Raffinose, and Melibiose indicate growth rate (×1000) in the medium with each of the sugars as the sole carbon source; Glu-Gal indicates difference between the growth rates in glucose and in galactose; *MAL31* and *GAL2* indicate CNV of each of the genes; *MEL1* indicates presence (solid bar) or absence (blank) of the gene; 40 °C, 41 °C, and 9% ethanol indicate growth efficiencies under the pressures

maltose, they harbor duplicated *MAL31*, *MAL32*, and *MAL33* genes as the other domesticated isolates do. Correspondingly, they also share elevated maltose utilization ability with the solid-state fermentation group with maltose as one of dominant carbon sources in their living environments (Fig. 5). These data imply that the two major groups of domesticated isolates originate from a common origin with elevated maltose utilization ability. The solid-state fermentation group containing isolates exclusively from traditional fermentation processes in Far East Asia is apparently native to this region. We therefore infer that the liquid-state fermentation group should also originate from the same region.

The Far East Asia origin hypothesis for the domesticated lineages recognized worldwide so far is inconsistent with a previous study[27] showing that wine isolates were domesticated first in Europe from the MO lineage of *S. cerevisiae*. However, our data suggest that it is more likely that the European wine isolates were transferred from Asia. First, the Wine lineage contains four Chinese isolates from fruit and orchards. Second, European Wine isolates share HGT genes (regions A and B) with Chinese wine and wild isolates (Fig. 4 and Supplementary Data 7). It is unlikely

that the Chinese wild isolates obtained the HGT fragment for European wine isolates, given the origin of the domesticated lineages from the wild and the general gene flow from the wild to the domesticated populations. Third, the close relationship between the Milk and Wine lineages also supports the origin of the latter from Asia. The phylogenetic trees (Fig. 1 and Supplementary Figs. 1 and 2) show that the Wine and the Milk lineages originated from a common ancestor. The sharing of duplicated *MAL* genes and elevated maltose utilization ability in these two lineages as discussed above also support their common origin. The milk isolates were all from traditionally fermented dairy products sampled from local families in remote pastoral areas covering western and northern China and Mongolia (Supplementary Fig. 4) and exhibited higher genetic diversity ($\pi = 3.86e-03$ Supplementary Data 3) than the Wine/European isolates ($\pi = 1.59e-03$ or less)[18,20,27]. The data suggest that the Milk lineage is native in Asia and originates from an Asian ancestor which is also shared by the Wine lineage. Though isolates in the two Beer lineages within the liquid-state fermentation group have not been sampled from Far East Asia, the beer-brewing history in China has been dated back to 5000 years ago[42],

much longer than the domestication history of beer yeasts in Europe (AD 1573–1604) as estimated previously[19].

The high degree of heterozygosity shared by almost all domesticated isolates and the homozygosity shared by all wild isolates are striking. The high level of heterozygosity in domesticated isolates of *S. cerevisiae* was also observed in previous studies[13,18,19,34], but the phenomenon was attributed to the ploidy level higher than $2n$[19] or to long periods of asexual reproduction[18]. However, we show here that the polyploidy variations in wild and domesticated isolates are similar. Our population structure analysis shows that recombination is more frequent in domesticated than in wild populations (Fig. 1b and Supplementary Fig. 1). Therefore, the heterozygosity shared by the domesticated lineages is unable to be explained by higher ploidy level and clonal reproduction alone. One alternative explanation is that the ancestor(s) of the domesticated lineages was/were formed by outcrossing between genetically different wild isolates, as hypothesized by Magwene[34,43]. The heterozygosity of the domesticated isolates is probably maintained due to the loss of sexuality, reduced spore viability and the advantage of heterosis[44] for living in nutrient rich fermentation environments.

Previous studies on population genetics and genomics of *S. cerevisiae* resulted in controversial answers to a fundamental question in biology concerning whether the diversity and evolution of organisms are primarily driven by natural selection or neutral genetic drift[9,17–20,22,23,25], echoing the long standing selectionist–neutralist debate. We show here that the genetic diversity of *S. cerevisiae* is mainly contributed by the highly structured wild population with greatly diverged lineages in China. However, neither geographic nor ecologic factors can explain the structure of the wild population. Wild isolates from the same locations may belong to greatly diverged lineages, exhibiting a phenomenon of sympatric differentiation. On the other hand, a single lineage may contain isolates from geographically well separated regions (Supplementary Data 1)[28]. These phenomena suggest that immigration and secondary contact of *S. cerevisiae* isolates from different lineages is probably common in nature. However, genetic admixture in the wild population has rarely been detected, suggesting reproductive isolation among different wild lineages is well established. The mechanism remains to be fully revealed. Previous studies[28,45,46] suggest that large-scale chromosomal rearrangements might play a role in the onset of reproductive isolation in *S. cerevisiae*. The lineage specific large alien fragments obtained through HGT and introgression in wild populations (Fig. 4 and Supplementary Data 7) may also cause chromosomal structure variations similar to chromosomal rearrangements and probably also contribute to reproductive isolation. We also find that lineage specific CNVs (Fig. 3 and Supplementary Fig. 9) and positive and purifying selection (Supplementary Data 9 and Supplementary Note 5) are rare in the wild population. Our results are consistent with a neutral model for the evolution of the wild population of *S. cerevisiae*.

On the other hand, the domesticated population of *S. cerevisiae* is apparently an outcome of adaptive evolution in the life history of the species. Our results suggest that the domesticated populations of *S. cerevisiae* were probably formed through a single bottleneck leading to the creation of heterozygous offsprings that adapted to nutrient-rich environments, specifically to a maltose-rich environment at the beginning. The extensive gene expansion and contraction leading to consequent phenotypic trait changes in domesticated populations indicate adaptive evolution due to strong selection for specific niches. Many genes associated with stress resistance, environment response and sugar transportation and metabolism are generally duplicated in domesticated lineages, while a considerable number of other genes unnecessary in the

nutrient-rich fermentation environment, including four FLO genes, are contracted or lost (Fig. 3 and Supplementary Data 6). The maintenance of the FLO genes in the wild lineages suggests that cell adherence and biofilm formation are important for the yeast to survive in the wild. However, in nutrition rich fermentation environments, especially in solid-state fermentation and even in liquid-state fermentation processes without a yeast cell separation step (e.g., dairy product fermentation), this trait may not be required. In contrast, planktonic cells may have an advantage for their proliferation in fermentation environments, especially in spontaneous fermentation processes with other microbes. Further comparative studies on flocculation of wild and domesticated isolates will be needed to test this hypothesis. The enhanced flocculation ability of some industrial isolates for beer, wine, and bioethanol production[47], is apparently a post-domestication trait arising due to strong artificial selection for cell separation after liquid-state fermentation. These fermentation processes usually use elaborately bred pure yeast cultures.

Ecology seems to be the primary force driving the diversification within the domesticated population of *S. cerevisiae*. Domesticated isolates associated with different fermentation environments usually formed distinct lineages, suggesting genetic differentiation due to niche adaptation. Indeed, we observed remarkable lineage specific genomic and phenotypic variations in the domesticated populations, most remarkably in the Milk lineage (Fig. 3, Supplementary Fig. 9, and Supplementary Data 6), which possesses a duplicated *GAL2* gene and the introgressed *GAL7-GAL10-GAL1* gene cluster, resulting in an elevated galactose utilization rate (Fig. 5 and Supplementary Data 8). The contribution and mechanism of the introgressed *GAL* gene cluster to the elevation of galactose utilization rate beyond even the glucose utilization rate in the Milk lineage remain to be illuminated.

## Methods

**S. cerevisiae isolates**. A total of 266 *S. cerevisiae* isolates were employed in this study, including 106 wild and 160 fermentation-associated isolates. The wild set consists of 94 wild isolates that we compared previously[28] and 12 new ones isolated from primeval forests located in central, southeast and southwest China. The wild strains were isolated using the enrichment method[28]. The fermentation-associated set consists of 150 isolates associated with spontaneous fermentation of various traditional foods all over China and ten isolates used for commercial active dry yeast cell products available in the market in China. The fermentation-associated yeast strains were isolated using the dilution plating method and the yeast extract–peptone–dextrose (YPD) agar (w/v, 1% yeast extract, 2% peptone, 2% D-glucose, and 2% agar) supplemented with 200 μg/mL chloramphenicol. Yeast isolates were identified as described previously[28].

**Phenotypic characterization**. The mitotic proliferative abilities of the isolates in liquid synthetic defined medium (0.67% Yeast Nitrogen Base, Difco) with 2% glucose, galactose, sucrose, maltose, melibiose and raffinose, respectively, as the sole carbon source at 30 °C; in liquid YPD (1% yeast extract, 2% peptone, and 2% glucose) with 9% ethanol at 30 °C; and in liquid YPD at 40 °C and 41 °C, respectively, were tested in duplicate in microplates using a Bioscreen analyser C (Thermic Labsystems Oy, Finland) as described in Warringer and Blomberg[48]. The fitness variables including growth rate (population doubling time), lag (population adaptation time), and efficiency (total change in population density) were extracted from high-density growth curves and $\log_2$ transformed as described previously[22, 49].

**Sporulation and flow cytometry analyses**. The sporulation efficiencies was tested under the optimum conditions as formulated by Codón et al.[50]. Sporulation efficiency was calculated as the ratio between the number of sporulated cells (with 2, 3, or 4 spores) in several random sights and the total number of cells in the same sights. Spore viability was tested by dissecting at least 25 tetrads (100 spores) per isolates as described in Wang et al.[28] using the MSM 400 microscope platform (Singer Instruments, UK). Flow cytometry analysis was performed following the protocol as described in Albertin et al.[51] using a BD FACSCalibur flow cytometer (Becton-Dickinson, San José, CA, USA). *S. cerevisiae* strains FY1067 (diploid) and FY1067-01B (haploid) from the EUROSCARF yeast strain collection were used as calibration references.

**Genome resequencing, assembly, and annotation**. The genome DNA of each isolate was extracted using a standard Zymolyase protocol[52]. For the majority (259) of the isolates employed, a paired-end library with an average insert size of 300 bp was prepared and was sequenced using the Illumina Hiseq 2000 platform with 2 × 100 bp reads. The sequence coverages ranged from 68x to 439x (average = 193x; median = 190x). For the remaining seven isolates, which represent wild lineages from primeval forests (four isolates), original secondary forests (one isolate), and fermentation-associated isolates (two isolates), four DNA libraries including a mate-pair library with 3000 bp insert size and three paired-end libraries with 170, 500, and 800 bp insert size, respectively, were prepared and sequenced using the same platform to a coverage of 378x to 582x (average = 484x; median = 461x).

Raw reads were trimmed to remove low-quality (phred score ≤ 10), ambiguous and adaptor bases using the FASTX-Toolkit v0.0.14 (http://hannonlab.cshl.edu/fastx_toolkit/index.html). Then the program HiTEC[53] was used to correct reads and the error rate was set to 0.01. The unpaired reads were removed. The programs Velvet v1.2.10[54] and ABySS v1.9.0[55] were used to assemble clean reads. The hash value for Velvet and the parameter "k" representing the k-mer length for ABySS were optimized for each isolate. The parameter "-cov_cutoff" of Velvet and the parameter "n" of ABySS, corresponding to the minimum coverage nodesduring assembly progress, were adjusted for each sample. The assemblies obtained from Velvet and ABySS for each isolate were compared and the assembly with the longer contigs and a higher N50 was selected for further analysis. For the majority (255) of the isolates sequenced, Velvet achieved better assemblies, while for the remaining 11 isolates (AFB.1, WJZ1.2, ANG1, GS6, JM28.13, SXJM4.1, WL1, JM8.3, XST, HN5.1, and LJM21.3), the assemblies from ABySS were better. Pilon v1.16[56] was then used to improve draft genome assemblies by correcting bases, fixing misassemblies and filling gaps relying on the k-mer value determined by the hash value of Velvet or the parameter "k" of ABySS. PAGIT v1.01[57] was applied for improving further the quality of genome assemblies from Pilon by correcting base errors and closing gaps in consensus sequences and assembling contigs into chromosomes using the reference genome of S. cerevisiae S288c.

AUGUSTUS v2.5.5[58] was employed for gene prediction from the final assemblies generated by PAGIT with S. cerevisiae S288c as the model using the following parameters (genemodel = complete, protein = on). Then the BLAST program[59] was used to annotate the gene function through searching for homologous sequences in the Saccharomyces Genome Database (SGD) and GenBank. Based on in-house perl scripts, the genes recognized in each new genome assembly were coordinated with those of S. cerevisiae S288c. When a gene is described as "putative protein of unknown function" or "dubious open reading frame" in the SGD database, it is regarded as "a gene with unknown function" in this study.

**Reference-based alignment and variant calling**. The clean paired reads obtained were mapped to the S288c (R64-1-1) genome using the Bowtie2 program[60] with default settings. SAMTools v1.3[61] was employed to convert the alignment results into the BAM format and Picard Tools v1.56 (http://picard.source-forge.net) and BCFTools v0.1.19 (http://www.htslib.org/doc/bcftools.html) were used to remove duplicated sequences. Finally custom Perl scripts were applied for extracting the variant bases. For these programs, following parameters were used: the maximum number of reads for calling a SNP = 10,000; the minimum mapping quality = 25; and the minimum base quality to identify putative SNPs = 25. In addition, the Genome Analysis Toolkit (GATK v2.7.2)[62] program was used to detect the variable sites. The parameters "stand_call_conf" (thresholds for low and high quality variation loci) and stand_emit_conf (minimum phred-scaled confidence threshold) were set to 40.0 and 30.0, respectively. The high-quality SNPs extracted were the consistent variation sites obtained from SAMTools and GATK. The variation sites with a coverage depth ≥15 were remained for subsequent analyses and final SNP extraction. The variation sites of an isolate with a coverage depth greater than four times of the sequence depth of the isolate were probably resulted from sequencing errors or duplicate sequences and thus were removed according to Lam et al.[63]. The variation sites were kept only when at least 80% of the reads were positive for homogeneous sites and at least 20% of the reads were positive for heterogeneous sites[64]. Finally, a total of 923,479 sites were extracted from the 266 isolates sequenced. The SnpEff v4.3i tool[65] based on the interval forest method was used to annotate and predict the effects of SNPs on genes.

In order to extract genome scale SNPs from the combined collection containing the 266 isolates sequenced in this study and 287 isolates sequenced previously in Liti et al.[20], Strope et al.[17], and Gallone et al.[18], the genome assembly of each of the previously sequenced strains were retrieved from the SGRP website or GenBank. The assemblies were mapped to the reference genome of S288c using BLAST search with thresholds of 75% nucleotide identity and 60% nucleotide coverage. Pairwise alignments of the sequences of the strain compared with the reference sequences of S288c were generated using the MAFFT program[66]. The bases at the variation sites from the reference sequences coordinating with the positions of SNPs found in the Chinese samples were extracted from the strain compared. For the sites where SNPs occurred in the Chinese isolates but were unknown or missing in the strain compared, these sites were treated as "N". A total of 783,440, 852,873, and 884,040 SNPs coordinating with the SNPs found in the Chinese isolates were extracted for the sets of strains sequenced in Liti et al.[20], Strope et al.[17], and Gallone et al.[18], respectively, when setting the "N" base threshold for each site to 10%.

Finally, a dataset of 736,689 SNPs at the consistent positions among the SNPs identified from different sets of isolates were extracted for the combined 554 isolate collection for integrated phylogenetic population structure analyses.

Similarly, a dataset containing 628 isolates including the MO isolates employed in Almeida et al.[27]. We selected 51 oak isolates, 21 fermentation isolates (16 wine, 2 beer, and 3 sake isolates) and 8 fruit isolates from the isolates sequenced in Almeida et al.[27] and abstracted 217,727 SNPs. Then, the bases at the variation sites from the reference sequences coordinating with the positions of SNPs found in the MO isolates were extracted from Chinese samples (217,521 SNPs), strains sequenced in Liti et al. (2009) (217,108 SNPs), Strope et al. (2015) (210,300 SNPs) and Gallone et al. (2016) (214,793 SNPs). Five isolates in Strope et al. (2015) with poor assemble genomes were excluded. Finally, a dataset consisted of 206,810 SNPs covering 628 isolates was constructed for subsequent phylogenomics and population structure analyses.

**Phylogenomics, structure, and population genetics analyses**. Phylogenetic trees were constructed based on the whole genome SNPs, including both homozygous and heterozygous sites, with the latter being encoded as the International Union of Pure and Applied Chemistry (IUPAC) ambiguity codes. The sequence alignment was subjected to maximum likelihood analysis using RAxML v8.0.0 with the GTRMMA model and bootstrap resampling was set to 100[67]. The repeated random haplotype sampling (RRHS) strategy with 100 repetitions was applied as described in Lischer et al.[68]. The 100 maximum likelihood trees generated were then summarized in a majority rule consensus tree with mean branch lengths and bootstrap values using the SumTrees program[69]. FastTree v2.1.3[70] with generalized time-reversible model was used to determine the topology of the phylogenetic tree. Population structure was inferred using the program ADMIXTURE v1.23[33], and the best-fit K value was determined by the cross-validation (CV) procedure of the program and the value with a minimum CV error was selected (Supplementary Fig. 3a, b). Principal component analysis for SNPs matrix was performed through the GCTA v1.26.0[71] program based on the binary format files generated by PLINK v1.07[72].

The nucleotide diversity ($\pi$, the average number of nucleotide differences per site) and the nucleotide polymorphism ($\theta$, the proportion of nucleotide sites that are expected to be polymorphic) of the collection of the 266 isolates and each population or group were calculated using the software Variscan v2.0.6[39] with the NumNuc parameter being adjusted for each group for including at least 80% of isolates within the group and parameters "CompleteDeletion = 0, FixNum = 0, RunMode = 12, and WindowType = 0" were selected.

Shared polymorphisms, fixed differences, and private polymorphisms in all groups that were considered were calculated using the EggLib tools[73]. The nonmissing data at a frequency smaller than 75%, and singletons were removed from the analysis. The IUPAC ambiguity codes were considered as the valid data. Other characters were treated as the missing data. Besides, for the IUPAC ambiguity sequence, each site was allowed for multiple mutations (alleles > 2). For the SNP matrix excluding heterozygous sites, the valid data was the four bases (A, C, G, and T), and each site was biallelic markers (alleles < = 2).

**Demographic history analysis**. We performed the demographic analysis using the software package ∂a∂i v1. 2. 3[74] as in Branco et al.[75] and Almeida et al.[27], based on the folded joint-allele frequency of the noncoding SNPs with minor allele frequency ≥0.01 in all populations considered. The noncoding regions were selected as described in Almeida et al.[27]. S. cerevisiae strain S288c (SGD release R64-1-1of 2011-02-03) was used as the reference genome. S. paradoxus (Y-17217) was used as the outgroup genome. In order to decrease the bias and improve the efficiency, we selected 97,895 noncoding SNPs from 32 wild and 36 domesticated isolates representing all the wild and domesticated lineages recognized in this study for demographic analysis. The candidate models split_mig (split into two populations of specified size, with migration), Isolation-with-Migration (IM) model with exponential growth, prior_onegrow_mig (model with exponential growth, split, bottleneck in domesticated group, population recovery, and migration) and prior_onegrow_nomig (model with exponential growth, split, bottleneck in domesticated group, population recovery, and no migration) were tested. Each model was run five times from independent starting values. Conventional bootstrapping (100 replicates) was performed for estimating convergent parameters. The result suggested that the IM model was the best fitting model for estimating the demographic parameters. The fractions of the ancestral population expressed as the estimated effective population size ($N_A$) that entered into the wild and the domesticated group were 99.36% and 0.64%, respectively. The migration from the wild to the domesticated group (0.5254) is significantly higher than (seven times) that (0.0723) of the migration from the domesticated to the wild group. When we selected lineages CHN-X, CHN-V, and CHNVI-VII representing the wild population, and ADY, Milk, Mantou1, Mantou6, and Qingkejiu lineages representing the domesticated population and repeated demographic analysis, we obtained similar results. The data suggest a strong bottleneck during the domestication history of yeast (Supplementary Fig. 6b).

**CNV analysis of genes**. CNVs were identified by mapping the clean reads to the S288c reference genome using SMALT v0.7.6 (Wellcome Trust Sanger Institute, https://sourceforge.net/projects/smalt/) with default parameters, except that the

step size and the k-mer value were set to 2 and 13, respectively. The mapping quality was set to 30 (p < 0.001) using SAMtools and PCR duplications were removed.

Regions of the genome showing CNV between isolates were identified with the Splint script to avoid the "smiley pattern" bias as described in Gallone et al.[18]. CNVs were detected in 1000 bp nonoverlapping frames with default internal parameters. Based on the result of CNV detection, the copy number of each frame was calculated as the ratio between pdepth and neutralpred, but 12 isolates with exceptionally biased Splint results were excluded in further analyses. Based on the CNV value of each frame, the median CNV value of each base in an open reading frame was used to determine the CNV of each gene. The copy number value of each gene was detected based on a discontinuous spline regression technique and a hidden Markov model, which can represent the relative values of CNVs. In order to acquire the optimal bins of the continuous CNV values, we analyzed the distribution of the CNV value of each gene among the 254 isolates compared. Depending on the relative CNV values, the degree of expansion or contraction of each gene was classified into different levels as shown in Fig. 3 and Supplementary Data 6 according to the following criteria: (1) complete deletion (CNV = 0) when the value is less than 1% CNV left tail (0.34); (2) partial deletion (CNV = 0.5) when the value is between 1% and 5% CNV left tail (0.73); (3) normal level (CNV = 1) when the value is between 5% CNV left tail to 5% CNV right tail (1.2); (4) two fold duplication (CNV = 2) when the value is between 5% and 1% CNV right tail (1.74); and (5) three or more fold duplication (CNV ≥ 3) when the value is more than 1% CNV right tail.

In order to map the phenotypic variation in maltose and galactose utilization with genomic changes, we estimated the copy numbers of genes in the MAL3x cluster on chromosome II, the MAL1x cluster on chromosome VII, and GAL2 on chromosome XII using the cn.Mops bioconductor package, which can reduce noise through Poisson distributions[76]. The GAL2 gene of the isolates from fermented milk products were introgressed from an unknown source and showed significant sequence divergence from those in the reference S288c genome. Therefore, the GAL2 sequence of S288c was replaced by the GAL2 sequence of a Milk isolate and the CNVs of this gene in the isolates associated with fermented dairy products were analyzed further. Copy numbers were calculated and normalized for 1000 bp windows. The cn.Mops package was unable to manipulate the SAM files of 266 isolates containing the whole genome information because of its limitation in memory use, the sequences covering only the three chromosomes II, VII, and XII containing the MAL and GAL genes were analyzed using the cn.Mops package with default parameters.

**Ploidy variation analysis.** In ploidy and chromosomal CNV analysis, considering the high extent of deviation of small or middle-large fragments, the median value of the 1000 bp nonoverlapping frames was used to calculate the CNV of each chromosome in an isolate. The original CNV value (Vo) of a chromosome determined from the genome sequence reads was then adjusted to the actual CNV value (Va) according to the relative DNA content value (D) of the isolate estimated from the result of flow cytometry analysis (Supplementary Data 5). The equation is: Va = D x (Vo−1). Based on the dispersion analysis of the actual CNV value (Va) of every chromosome in all the 266 isolates compared, the copy numbers of individual chromosomes in a specific isolate was estimated according to the following criteria: when Va is less than −0.7, one copy of the chromosome is missing; when Va is between −0.6 and 0.5, no deletion or duplication of the chromosome occurs; when Va is between 0.6 and 1.6, one extra copy of the chromosome exists; when Va = 1.7 − 2.6, two extra copies of the chromosome exist.

Note that three domesticated isolates (HN2.2 between lineages Qingkejiu and Mantou5; HQ3.1 in lineage Huangjiu; and GS3.1 between Mantou 3 and 4 in the tree) with relative DNA contents of 1.15–1.40 were identified as haploid isolates but showing signals of heterozygosity. They are probably not real haploidy. The problem was probably due to PI staining or FACS measurement errors.

**Introgression and HGT analyses.** If only one (e.g., S288c) or limited number of S. cerevisiae strains are used as references, it will be unable to detect the potential HGT or introgression events when the genes or fragments from other isolates have no homologs in the reference strains. To avoid this bias, we constructed a reference genomic library including the genome sequences of S. cerevisiae strains S288c, EC1118, FostersO, T7, and the 38 S. cerevisiae strains sequenced in Liti et al.[20] other Saccharomyces species including S. paradoxus Y-17217, Saccharomyces castellii Y-12630, Saccharomyces pastorianus Weihenstephan 34-70, S. mikatae IFO 1815, S. kudriavzevii ZP591 and IFO1802, S. bayanus 623-6C, Saccharomyces eubayanus FM1318 and CBS12357, and S. uvarum CBS7001; and species closely related with Saccharomyces including Lachancea kluyveri Y-12651, Kluyveromyces lactis Y-1140, Naumovozyma castellii CBS4309 and Y-12630, Zygosaccharomyces rouxii CBS732 and NBRC1876, and Torulaspora delbrueckii CBS1146 and Y-50541. The S. cerevisiae reference strains were used to detect potential introgression or HGT fragments and the other species were used to judge possible donors of the alien fragments in S. cerevisiae. The maximum genome sequence divergence between different lineages of S. cerevisiae is <1.7% (Supplementary Data 3). Thus, a gene or fragment with less than 95% sequence identity with its homolog in reference S. cerevisiae strains is generally considered as possible introgression or HGT events[17].

We identified 246 fragments with a minimum length of 1 kb when we set a threshold at 95% sequence identity. Then, we raised the threshold of identity and fragment length to 93% and 1.5 kb, respectively and removed short fragments occurring only in a single isolate. Finally, we identified 79 putative HGT or introgression fragments from the 266 isolates sequenced in this study. The de novo assembly sequence of each potential introgression or HGT fragment were split into 1000 bp frames and searched homologous sequences in the reference genomic library constructed using BLASTN (v2.5.0) by setting the window size and sliding window size to 1000 and 500 bp, respectively. When the sequence identity was less than 65% and the alignment coverage was less than 30%, the frame was regarded as deletion (the identity value was set to 0 in Supplementary Data 7). A total of 180 genes were identified from these potential introgression or HGT sequences, including 24 of the 34 genes encompassed in the three HGT regions found in the wine strain EC1118[10]. Individual gene sequences harbored in a putative HGT or introgression fragment were used in BLAST search through the NCBI sequence database and phylogenetic analyses to estimate or confirm the origin of the fragment as shown in Supplementary Fig. 10.

**Statistical analyses.** Standard statistical analyses were conducted in R project (v 3.3.1) (https://ww- w.rproject.org/) with custom scripts under available packages in the project. To acquire the gene list in Supplementary Data 5 and Fig. 3 with significant CNV difference (P < 0.01) between the wild and domesticated populations or different lineages, Student's t test or Wilcoxon test was performed for the dataset from two groups. For clustering analysis, we selected the furthest neighbor method to display the result of CNV dataset. We used Chi-squared test for correlation analysis of two class variables, such as the aneuploid ratio between the wild and domesticated populations, when each frequency of the two-way contingency table was more than five; otherwise Fisher exact test was used.

**Data availability.** The whole genome sequence data in this study has been deposited at DDBJ/ENA/GenBank under the Bioproject ID, PRJNA396809. The Biosample ID, SAMN07436807-SAMN07437072, and the Genome Accession numbers, NPOV00000000-NPZA00000000 are listed in Supplementary Data 2.

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

## Acknowledgements

We thank Heping Zhang, Inner Mongolia Agricultural University, for providing yeast isolates from fermented dairy products and Timothy James, the University of Michigan, for his suggestions and language edition. This study was supported by Grants 31470150, 91131004, and 31461143027 from the National Natural Science Foundation of China (NSFC) and QYZDJ-SSW-SMC013 from the Chinese Academy of Sciences.

## Author contributions

F.-Y.B. and Q.-M.W. conceived and designed the project. F.-Y.B., Q.-M.W., and P.-J.H. performed sampling and yeast isolation and identification. P.-J.H., W.-Q.L., and J.-Y.S. phenotypic characterization. S.-F.D., W.-Q.L., and J.-Y.S. performed sporulation and flow cytometry analyses. S.-F.D., K.L., and X.-L.Z. performed bioinformatics analyses. F.-Y.B. and S.-F.D. analyzed the data and wrote the paper.

## Additional information

**Competing interests:** The authors declare no competing interests.

