## [Peer Review File · Nature Communications]

Reviewers' comments:

Reviewer #1 (Remarks to the Author):

Duan et al. characterised the genetic and phenotypic diversity of a large collection of *Saccharomyces cerevisiae* strains isolated from wild populations and traditional fermentations from China/Far East Asia. In a previous study (Wang et al. 2012), these authors showed how Asia harbours the highest genetic diversity of wild *S. cerevisiae* strains, suggesting that China/East Asia is likely the origin center of *S. cerevisiae*. In this study, the authors report the presence in Asia of a domesticated population associated with traditional fermentations such as Mantou, Daqu, Baijiu, Huangjiu and Qingkejiu. The authors suggest that the domesticated population originated from a single ancestor adapted to maltose-rich environment and further diverged in two major groups differentially adapted to solid and liquid state fermentations.

Overall, the study provides a lot of interesting data and has the potential to really impact the field. However, I feel that there are several problems with the methodologies and analyses applied that need to be resolved. Moreover, some claims in the paper are insufficiently supported by data or inadequately discussed. Below, I've listed our main comments regarding (i) methodology and (ii) content/discussion.

Major comments:

(i) Methodology

1- Line 135 – Population structure analysis. The authors should add a supplementary figure including lower K_s and the cross-validation values. In addition, they should justify the selection of K_s from 20 to 27 that implies a strong assumption on a minimum number of ancestral populations of 20.

2- Line 145 – Linkage disequilibrium. It's hard to understand how the linkage disequilibrium analysis has been performed. For diploid, polyploid/aneuploidy individuals, like the isolates included in this study, it's possible to determine the genotypes but not the haplotypes. Consequently, we do not know which nucleotides reside together on individual chromosomes. For these reasons, the correlation coefficient (r^2) used to estimate linkage disequilibrium and its decay might be incorrect and should include an estimate of error due to the unknown gametic phase or this analysis should not be used to draw crucial conclusions.

3- Line 152. Bottleneck claim. The authors suggest the presence of a bottleneck in the evolutionary history of the domesticated lineages from the wild populations of *S. cerevisiae*. The presence of this bottleneck is not proven and the decrease in genetic diversity represents only a hint. Showing the presence of bottlenecks in yeasts is notoriously difficult because it's hard to estimate demographic parameters (unknown generation time, unknown haplotypes, unknown effective population size). Methods like the Approximate Bayesian computation (ABC) have the potential to infer some of these parameters with complex models but the final outcome might not be easily interpretable due to the specification a sub-optimal model that doesn't exhaustively represents the dataset. For a discussion regarding this topic start from Robinson et al., 2014 in *Molecular Ecology*.

4- Figure 1A. Phylogenetic reconstruction. I have two problems with this analysis. First, the authors use the method described in Xu et al. 2012, where evolutionary distances are first measured based on p-distance and the resulting distance matrix is used to construct a neighbour-joining tree. I disagree with the distance measure applied for homozygous and heterozygous genotypes described in this paper, and I don't think it is a good methodology. Second, if I understand the supplemental

information correctly, this method was also used in a wrong way by the authors. The authors cannot consider a heterozygous site T/G as TG in the alignment because in ML models each character is assumed to be independent, which is not the case for heterozygous sites. In addition, it is not clear whether the same method was applied for the phylogeny in Figure S1; can the authors provide information in the supplementary methods on how this phylogeny was obtained? Also, the position of the outgroup in the phylogeny in Figure S1 should be included.

If the authors want to attempt a phylogenetic inference considering heterozygous sites, RAxML can deal with heterozygosity coded as ambiguities codes, but the models are still quite experimental. If the amount of heterozygosity is high and due to recombination/reticulation a network approach would be more useful. In addition, using only SNPs alignments, omitting invariant positions, can lead to overestimation of evolutionary divergence and the model should include an ascertainment bias correction (included in RAxML). Please see discussions in (Leache' et al., 2015 in Systematic Biology; Lischer et al., 2013 in Molecular Biology and Evolution; the new RAxML manual)

5- Divergence time estimations. The phylogenetic strategy that is applied (with which I don't agree, see comment above), probably yields a correct topology, but definitely has a strong effect on the branch lengths estimation, which in turn has huge implications for divergence time estimates. Therefore, these calculations might not be correct. Also, the authors provide very little detail about which method was applied to calculate the divergence time, which molecular clock was used and why the calibration point for the split *S.cerevisiae* –*S.paradoxus* was set from an interval of 10 to 20 Mya (that is already very rough and based only on mutation rate) to an interval of 4 to 10 Mya. Also, it is not clear which mutation rate and generation time the authors considered, and whether they applied different parameters to domesticated vs wild populations. Or were the domestic populations completely discarded for the full analysis? Please give more details.

6- In Figure S6, phylogenetic tree (a), did the authors use *S. castellii* to root the tree? The placement of *K. lactis* vs *S. castellii* is wrong. *S. castellii* is a post WGD species and *Kluyveromyces* splitted from *Saccharomyces* before WGD. Furthermore, other trees look problematic from this point of view (cfr. placement of *Zygosaccharomyces* vs *Torulospora*). Make sure that you are not using divergent paralogs within each species that might wrongly place the species in the tree. Also make sure to add *Candida glabrata* to the analysis. In addition, it would be useful to include in the methods a section describing how the trees were obtained. Because of the short length of these fragments, it might be difficult to recapitulate the correct genealogy (although the divergence is quite high and this problem should be limited) but at least you should discuss the incorrect placements in the text.

(ii) Content

1- It would be useful to provide more details on the production process of the traditional Asian fermentations mentioned in the manuscript (including at which step fermentation happens and if and how there is re-use of the fermentation product to inoculate a new batch, at which temperature fermentations are usually performed, what ethanol concentration they reach, etc). Also, which of these are industrial processes, and which are small, traditional practices performed by local farmers? This would be especially relevant information for the strains in the 'Milk' lineage. Figure S1 shows that these strains are in a sister lineage of the Wine strains, and thus must have diverged from the Wine clade AFTER they split from eg. the Beer 1 clade. Therefore, it is intuitively more likely that these Milk strains are more commercial/industrial strains that were initially domesticated in Europe and later transferred back to Asia, no? Or how do the authors interpret this phylogeny? Additional information on the life history of the strains and details on the specific fermentation processes would greatly help to interpret the genetic (and even the phenotypic) data.

2- As many people are not familiar with Chinese geography, it would be useful to provide as a

supplemental figure a map with the geographical origin of each strains. The presence of geographical substructure might have a strong effect on results regarding the diversification of domesticated vs wild strains, or diversification of the different wild lineages. E.g. in the PCA plot (Fig S3), domesticated strains might not separate into different subpopulations as observed for wild strains, because they have been sampled from a smaller area? Moreover, this information could serve as a tool to hypothesize about the initial region where strains were first domesticated?

3- Throughout the manuscript it's difficult to interpret the major conclusions regarding to origin of domesticated lineages. It would be interesting if the authors would discuss the relationship between the newly identified domesticated lineages in this study and the domesticated lineages identified in previous studies (Liti et al. 2009, Gallone et al., 2016, Goncavales et al., 2016) in more detail. Are the authors implying that all the domesticated lineages originated from a common DOMESTICATED ancestor in Asia that was subsequently spread in Europe and further diversified in the e.g. Wine/Beer lineages? Or rather that WILD cerevisiae strains travelled from Asia to Europe, and were domesticated there? Especially the placement of the Milk strains in Figure S1 is an interesting observation in this regard. The first option would go against the hypotheses of Almeida et al. 2015, who identified a wild stock of yeast that they believe are the wild genetic stock of domesticated wine yeasts. The second option also implies that *S. cerevisiae* has been domesticated multiple times in different geographical locations (Europe/Asia), which is maybe more intuitive? Nevertheless, this should be stated more clearly and discussed in more detail.

4- Line 224-231. The MK test. How many isolates of *S. paradoxus* have been used in the MK test? Also, can the authors expand the discussion regarding their MK test? It would be interesting to discuss the higher number of genes in purifying selection detected in the domesticated versus wild populations in more detail. How can this signal be linked to differences in population dynamics (expansion vs bottleneck), mating behaviour (sex vs no-sex) and heterozygosity? The way this result has been added to the text so far doesn't have any additional value. More information on the history of these traditional fermentation processes (like mentioned earlier) might help the interpretation of such results. For instance, are these strains continuously growing in rich medium, fluctuating environments, is it season-bound, is there back-slopping?

5- Figure 4. Introgression analysis. The database to which the fragments are blasted is very limited and the results are therefore misleading. E.g. fragment 1 in table S6: this fragment is present in CHN-IX lineage with 100% identity percentage across strains. When you blast it against your custom database your top hit for 2/3 of the length of the fragment is *K. lactis*. But, it matches for 71%. Therefore, this fragment is obviously not a fragment from *K. lactis* and reporting it as top match is very misleading even if you report it as closely related to *K. lactis*. The authors should vastly extend their custom database and look for better top matches or you have to report this fragment as unknown source. In addition, in Figure 4 legend (top right) the percentage cut-offs are confusing and sound a bit arbitrary (e.g. *K.lactis* is reported only once – pink line on the top of the graph – for a hit that in the table has 84% identity, while in the legend it states '*K. lactis* < 90%').

6- Line 203-210 and discussion. The discussion regarding FLO genes and the correlation to the flocculation phenotype for the strains analysed, sounds a bit out of place because flocculation has not been assessed. Moreover, while the authors argue that flocculation is not important in industrial strains, it is of huge importance in eg. beer brewing, bioethanol production and winemaking (eg. sherry). In addition, the phenotypic variability observed for this phenotype is not necessarily linked to changes in copy number, but it is known that there are huge differences in activity of the different gene variants and their expression. In addition, it should be noted that the FLO genes are not only responsible for flocculation, but also are involved in pseudohyphal growth, a potential survival strategy for wild yeasts (see the paper by Magwene et al where they hypothesize that sporulation and

pseudohyphal growth are two survival strategies for wild yeast). It would be interesting if the authors would measure flocculation and pseudohyphal growth to strengthen the conclusions of their genetic analysis, and possibly even hypothesize on the role of these traits in wild/domesticated strains.

7- Line 332-336. The single origin hypothesis is not really supported, especially not by point 1 and 2 (or at least not by the limited information and discussion that is given for these points). For point 3, common expansion and contraction of certain genes, e.g. maltose, might indicate the presence of one maltose adapted ancestor but does not exclude the presence of multiple ancestors that adapted to maltose, especially if this statement is based on the CNV of only one single maltose gene Mal31, present in highly variable regions (subtelomeres).

8- Line 359-368. What's the number of private/shared polymorphisms within each domestic lineage? If the author's hypothesis regarding one single common ancestor derived from an outcross is true, many SNPs/heterozygous sites will be shared across the Asian (and European) domesticated lineages.

9- Line 385-396. The hypothesis on reproductive isolation of wild strains is interesting, but should be supported by experimental data. We suggest to select one representative from each wild lineage and perform crosses to test the hypothesis.

Minor comments:

Figure 1A/Figure S1. Phylogenetic reconstruction. Bootstrapping values or any other type of confidence values are missing for all the phylogenies reported in the text. Please provide confidence values.

Figure 2. Can you explain the heterozygosity for the haploid strains? Is this a PI staining error or rather sequencing errors?

Figure 3. It's very hard to gain any clear insight from this figure. Why not sorting the genes according to GO categories on the top (annotate which GO category) and have a last column with No GO category? Because some genes will be present in multiple GO categories, maybe you can report only major GO categories?

Figure 4. In general, this figure is difficult to follow and the meaning of each colour code should be clearer. For example, one way to remove a layer of complexity is to sort the fragments based on species assignment rather than the arbitrary numbering on the bottom. Also, is it surprising that some fragments are present in all the strains? Can these really be defined as introgressions?

Figure S4a. Can you replace the y axis name from heterogeneity to heterozygosity?

Figure S4b. Sporulation should be replaced to "Sporulation efficiency" on the y axis and converted to percentage

Figure S4c. Spore viability on y axis should be converted to percentage

Figure S6 : tree a/b *Saccharomyces castellii* is misspelled

The separation liquid/solid state fermentations is probably a bit misleading. Intuitively, this terminology implies that strains belonging to the two groups were specifically adapted to the two types of processes, while for example Chinese bioethanol strains are present in the solid state fermentation group, and eg. all European bread strains are in the 'liquid' group. Would it make more sense to call them 'Asian' vs 'European' domesticated groups? The only problem with that terminology is the unexpected placement of the Milk lineage in the tree as a sister clade of the Wine lineage (see also other comments).

It would be interesting to compare the nucleotide diversity for the different lineages measured in this study to those calculated for the other known domestic lineages and report how they compare.

Line 176. Ploidy measurements. Can you clarify in which way did you combine the ploidy measurements obtained from PI staining together with the sequencing coverage?

Line 273 – (The majority of these phenotypic variations are also clearly correlated with specific genomic variations). I think this statement is not sufficiently supported by data (only 3 examples; galactose, melibiose and maltose).

Line 291: Include statistics.

Line 299-300. The authors do not show any specific genome or gene changes associated with raffinose and sucrose utilization, tolerance to high temperature and 9% ethanol. This statement is not supported by data.

Line 303 . The authors should mention the arguments supporting the Gondwanan origin hypothesis and how their conclusions rather support a Far East Asian origin.

Line 330-331 – (...implying the possibility that the domestic lineages recognized so far originate from a single isolate that originally existed or formed in such man-made environments.). In this statement do the authors refer to the domesticated populations in China or all the domesticated populations recognized so far for *S. cerevisiae*?

Line 370 – (.advantage of heterosis for living in nutrient rich fermentation environments?). Can the authors elaborate this statement or give a citation?

The authors should check the numbering of supplementary tables in the text and in the figure legends. There is some mixing between table S6 and S7: two tables are labelled S6 (MK test and introgression)

Reviewer #2 (Remarks to the Author):

In this paper, the authors sequence 266 *Saccharomyces cerevisiae* isolates from diverse sources in China. These strains are mixed between domesticated and wild strains. Using their data, as well as comparison of their data to other recent resequencing datasets from *S. cerevisiae*, they obtain new insights into the origin and evolution of this important model organism that also play a key role in human food and drink production.

I enjoyed reading this paper. In my opinion, the methods were appropriate, the results were novel, and the conclusions were in line with the results.

Regarding my specific comments:

1. Although the text is clear, the wording could be tightened. As an example, in the summary alone, the first sentence probably doesn't need the word 'as' at the beginning, the second sentence says 'origin' twice, the third sentence is missing the word 'on' near its beginning, and the fourth sentence says 'probably' twice. This type of critical editing could be extended throughout the manuscript.

2. I felt the authors could get more succinctly consolidate past work on yeast population genomics and more quickly and effectively make their important point. Uncertainty regarding the origins of different yeast lineages and the role of selection vs. drift in ongoing evolution in this species reflect a sample issue. Specifically, China had not been sampled extensively and it turns out China is arguably the most important place for unraveling all these issues. The authors seem to try to imply this in the paragraph from lines 57 to 69, but they need to explicitly make their point. The authors need to just directly spell things out for the many readers who will have no familiarity with their research area.

3. In the section on gene expansion, contraction, etc., I am not sure why the FLO genes are classified as involved in stress response. These genes tend to be involved in multicellular phenotypes, which can, but don't have to, be induced by stress.

4. Genes with unknown function. This term is used in the main text, but I wondered if this referred to

lack of annotated function on SGD or something else. The reason I ask is Augustus was used for de novo annotation and then genes were BLASTed against different databases of known genes. It wasn't clear to me how many new genes were found relative to the reference genome or other existing *S. cerevisiae* genomes using these de novo annotations? Also, it wasn't clear how often these new genes had ascribed functions? Many genes with unknown functions exist in SGD, but this might not be what the authors meant. Probably the methods and supplement need to be built out a little along these lines, while in the main text it might better to say something more specific than 'genes with unknown functions.'

5. In Figure 3, it is a bit confusing that copy number is provided as a continuous range from 0 to greater than 3. I realize that noisy coverage data are used to infer copy number but is there some way to obtain a maximum likelihood estimate of the coverage of a given gene in a given strain. It is also a bit hard to look at because the coloring scheme employed has 8-ish levels and my eyes are drawn to the grey, which is the baseline coverage of 1. Anyways, I think the aesthetics of this figure could be improved to better emphasize the result.

6. In Table S6, the header 'neutral selection' is used. Obviously there is no such thing, so this needs to be modified. I wasn't sure what was meant by this, so I didn't suggest a replacement.

Reviewer #3 (Remarks to the Author):

This manuscript presents an analysis of genomes and phenotype data for new strains of *S. cerevisiae* to answer questions about domestication. The strains are from sources not previously examined (rice wine, steamed bread, distilled liquors – all from china) and provide significant and new insights into the origin and domestication of yeast strains. Overall the manuscript is well presented, thorough in the analysis and represents a large body of work and a strong advance. However, there are a number of key points that should be addressed. I think most of these can be addressed by more careful and conservative statements of the claims made but also inclusion of alternative models in the discussion.

1. Phylogenetic analysis is not appropriate for individuals in a population. Although prior studies continue to build trees and talk about distinct lineages, it is neither correct nor justified. *S. cerevisiae* strains mate and recombine with one another. We don't talk about humans as forming distinct lineages but rather distinct populations. That said it is acknowledged that trees can be useful ways of presenting data. However, the nomenclature and particularly the use of language like distinct lineage is misleading. These are populations and we know they are interbreeding.

2. Single recent ancestor of domesticated strains. The proposal that domesticated lineage arose from single isolate is not supported. The observations used to support this do not exclude other possibilities as well as known admixture/introgression events that support the notion of a domesticated population rather than a single isolate. Later statements, "the heterozygosity shared by almost all domestic lineages is unable to be explained by cloning reproduction". This I agree with and shouldn't have been posed in ref 18. However, taking one step away and suggesting that domesticated lineages arose from a single hybrid is similarly ill conceived, not just because the data are consistent with a model of domestication with outcrossing, but because many individual genes often show histories that are not consistent with a single origin.

3. The claim that liquid and solid domesticated yeast strains arose from a single ancestral domestication event is not well supported. However, there is strong evidence that there was a single event in the origin of domesticated strains from China presented in this study. However, this quite likely did not include european/wine strains. First, it doesn't account for wild mediterranean oak

populations. Second, not many wine strains were used in the analysis. The observations that the MAL genes are present in the milk and wine strains is certainly interesting and does point to some shared ancestry between the European wine and Asian domesticates, but it does not imply an origin of these strains in China.

4. The dates for the divergence time need to explicitly state what assumption is made, I would prefer generations/yr. The reference to Hittinger (2013) for 10-20 Mya is not good, this is a review and is based on calibrations in plants and animals and perhaps one fungal fossil over 100 Mya. Whatever the solution, either remove the dates or make explicit in the text (not supplement) that these numbers are quite questionable.

5. MK test results are misinterpreted. $NI > 1$ indicates $Pn/Ps > Dn/Ds$. Assuming syn. sites are neutral this means either Pn is inflated or Dn is deflated. Dn can be low due to purifying selection – but this is divergence between species, which is shared by both domestic and wild strains and so wouldn't explain a difference between the two. Pn can be inflated due to loss of selective constraint, i.e. weaker purifying selection. The result of reduced purifying selection in domesticated strains has been found previously in yeast, e.g. Elyashiv et al. 2010.

6. The beginning of the discussion, there should be a distinction between the Far East Origin of particular species as compared to the domesticated strains. It is fine to discuss both, but the present manuscript only deals with domesticated strains of *S. cerevisiae*. See these suggestions:

Line 302 "origin of some *Saccharomyces* yeasts"

Line 303 "was also proposed recently for other species"

Line 304 "provide evidence" ... origin hypothesis for *S. cerevisiae*"

Other comments

Line 19. "Based on genomic"

Line 58. "could be due to insufficient"

Line 60. "A recent" - the tone is too combative in this paragraph. I think these changes would help.

Line 66-67. Check Naumov's work, I believe he has made this hypothesis as well.

Line 134. remove "that consistent of"

Line 173. The negative correlation – it would be worth citing Magwene (2011) who had a similar result.

Line 208 identified rather than recognized.

Line 283 and other places discussing the Maltose result – Naumov has (I believe) stated this in past work that Maltose is a strong indicator of domestic vs wild yeasts.

Figure 1 – the structure analysis on the bottom is not useful since one can't tell which strains the bars refer to. The full info is in the supplement so the Fig 1 panel can be removed.

Responses to Reviewers' comments

Reviewer #1 (Remarks to the Author):

Duan et al. characterised the genetic and phenotypic diversity of a large collection of *Saccharomyces cerevisiae* strains isolated from wild populations and traditional fermentations from China/Far East Asia. In a previous study (Wang et al. 2012), these authors showed how Asia harbours the highest genetic diversity of wild *S. cerevisiae* strains, suggesting that China/East Asia is likely the origin center of *S. cerevisiae*. In this study, the authors report the presence in Asia of a domesticated population associated with traditional fermentations such as Mantou, Daqu, Baijiu, Huangjiu and Qingkejiu. The authors suggest that the domesticated population originated from a single ancestor adapted to maltose-rich environment and further diverged in two major groups differentially adapted to solid and liquid state fermentations.

Overall, the study provides a lot of interesting data and has the potential to really impact the field. However, I feel that there are several problems with the methodologies and analyses applied that need to be resolved. Moreover, some claims in the paper are insufficiently supported by data or inadequately discussed. Below, I've listed our main comments regarding (i) methodology and (ii) content/discussion.

>> Thanks for this generally positive comments.

Major comments:

(i) Methodology

1- Line 135 – Population structure analysis. The authors should add a supplementary figure including lower K s and the cross-validation values. In addition, they should justify the selection of K s from 20 to 27 that implies a strong assumption on a minimum number of ancestral populations of 20.

>> This is a misunderstanding that was probably caused by the long and complicated sentence in the last version. Actually we selected $K = 20$ for the dataset containing 266 isolates sequenced in this study and $K = 27$ for the dataset containing additional 287 isolates sequenced in previous studies (Liti et al., 2009; Strobe et al., 2015 and Gallone et al., 2016). The best fit K values were validated by the cross-validation (CV) test as shown in the newly added Figure S3. We also have provided supplementary figures to show the structures at $K = 18$ and 19 for the 266 strain dataset and at $K = 25$ and 26 for the 554 strain dataset in the new Figure S3. The paragraph in the main body (Lines 138-149) has been substantially revised and we hope it is clearer.

2- Line 145 – Linkage disequilibrium. It's hard to understand how the linkage disequilibrium analysis has been performed. For diploid, polyploid/aneuploidy individuals, like the isolates included in this study, it's possible to determine the genotypes but not the haplotypes. Consequently, we do not know which nucleotides reside together on individual chromosomes. For these reasons, the correlation coefficient (r^2) used to estimate linkage disequilibrium and its decay might be incorrect and should include an estimate of error due to the unknown gametic phase or this analysis should not be used to draw crucial conclusions.

>> It is true that the majority of the isolates we sequenced are diploid and it is hard to phase our SNP matrix. We used PLINK v1.07 for LD analysis which can estimate the squared correlation based on genotypic allele counts for unphased SNP matrix (Purcell et al., 2007). Although the result is not identical to the R-square as estimated from haplotype frequencies (phased SNP matrix), it will be very similar judged from the principle of the PLINK analysis. In addition, the wild isolates we studied are almost homozygous and thus the LD of the wild group would be close to the value based on the phased SNP matrix. The LD decay (with a half maximum at 2.8 kb) of the domestic group estimated in this study is similar with that (decaying to half its maximum value at 3 kb or less) showed in Liti et al. (2009) who used haploid strains. Therefore, we believe the result of our LD analysis is reliable. We only use the LD analysis to show that recombination is more frequent in the domestic population than in the wild population, being consistent with the structure analysis.

3- Line 152. Bottleneck claim. The authors suggest the presence of a bottleneck in the evolutionary history of the domesticated lineages from the wild populations of *S. cerevisiae*. The presence of this bottleneck is not proven and the decrease in genetic diversity represents only a hint. Showing the presence of bottlenecks in yeasts is notoriously difficult because it's hard to estimate demographic parameters (unknown generation time, unknown haplotypes, unknown effective population size). Methods like the Approximate Bayesian computation (ABC) have the potential to infer some of these parameters with complex models but the final outcome might not be easily interpretable due to the specification a sub-optimal model that doesn't exhaustively represents the dataset. For a discussion regarding this topic start from Robinson et al., 2014 in *Molecular Ecology*.

>> We agree that in the last version we estimated the presence of a bottleneck based on the observation of the clear separation of the domestic population from the wild population in China in the phylogeny tree, the reduced genetic diversity of the former calculated from nucleotide diversity (π and θ) and the tendency of gathering together of the domestic lineages in the PCA plot. Thanks to this review comment, we have tried to use the Approximate Bayesian Computation (ABC) method implemented in DIYABC

2.1.0 for inference on population history. Unfortunately, the program ran very slowly and exceeded the memory power of our computing system. We then performed the demographic analysis using the software package *∂a∂i* v1.2.3 (Gutenkunst et al. 2009) as in Branco et al. (2015) and Almeida et al. (2015), based on the folded joint-allele frequency of the non-coding SNPs with minor allele frequency ≥ 0.01 in all populations considered. In order to decrease the bias and improve the efficiency, we selected 97,895 non-coding SNPs from 32 wild and 36 domestic isolates representing all the wild and domestic lineages recognized in this study for demographic analysis. The candidate models *split_mig* (Split into two populations of specified size, with migration), *IM* (Isolation-with-Migration model with exponential growth), *prior_onegrow_mig* (Model with exponential growth, split, bottleneck in domestic group, population recovery, migration) and *prior_onegrow_nomig* (Model with exponential growth, split, bottleneck in domestic group, population recovery, no migration) were tested. Each model was run five times from independent starting values. Conventional bootstrapping (100 replicates) was performed for estimating convergent parameters. The result suggested that the *IM* model was the best model for estimating the demographic parameters (Fig. S7). The fraction of the ancestral population that entered into the wild group and the domestic group is 99.36 % and 0.64 %, respectively. The migration from the wild to the domestic group (0.5254) is significantly higher than (seven times) that (0.0723) of the migration from the domestic to the wild group. We think that the data suggest a strong bottleneck during the domestication history of yeast. We have provided the new data in the main body (Lines 167-174) with a new supplementary figure (Fig. S7) and the method has been added in the Supplementary Information (SI Lines 210-236).

4- Figure 1A. Phylogenetic reconstruction. I have two problems with this analysis. First, the authors use the method described in Xu et al. 2012, where evolutionary distances are first measured based on p-distance and the resulting distance matrix is used to construct a neighbour-joining tree. I disagree with the distance measure applied for homozygous and heterozygous genotypes described in this paper, and I don't think it is a good methodology. Second, if I understand the supplemental information correctly, this method was also used in a wrong way by the authors. The authors cannot consider a heterozygous site T/G as TG in the alignment because in ML models each character is assumed to be independent, which is not the case for heterozygous sites. In addition, it is not clear whether the same method was applied for the phylogeny in Figure S1; can the authors provide information in the supplementary methods on how this phylogeny was obtained? Also, the position of the outgroup in the phylogeny in Figure S1 should be included.

If the authors want to attempt a phylogenetic inference considering heterozygous sites, RAxML can deal with heterozygosity coded as ambiguities codes, but the models are still quite experimental. If the amount of heterozygosity is high and due to

recombination/reticulation a network approach would be more useful. In addition, using only SNPs alignments, omitting invariant positions, can lead to overestimation of evolutionary divergence and the model should include an ascertainment bias correction (included in RAxML). Please see discussions in (Leache' et al., 2015 in Systematic Biology; Lischer et al., 2013 in Molecular Biology and Evolution; the new RAxML manual)

>> Because of the limitation in calculation power, we followed Fay and Benavides (2005, PLoS Genetics) and Liti et al. (2009, Nature) and many others in phylogenetic and phylogenomic analyses using distance based algorithms (e.g. neighbour-joining) based on extremely large datasets. For counting for heterozygosity, we followed the MLST analyses of *Candida albicans* in the treatment of heterozygous sites (e.g., Tavanti et al. 2005, J Clin Microbiol; Odds et al., 2007, Eukaryot Cell; Ge et al., 2012, Fungal Genet Biol) in the last version.

In order to accommodate this review comment, we re-performed phylogenetic analyses using the Maximum Likelihood algorithm implemented in RAxML and followed Lischer et al. (2013) in the treatment of heterozygous sites. We encoded heterozygous sites as IUPAC ambiguity codes and applied the repeated random haplotype sampling (RRHS) strategy with 100 repetitions. The 100 ML trees generated were then summarized in a majority rule consensus tree with mean branch lengths and bootstrap values using the SumTrees program (Sukumaran and Mark, 2010). The new ML trees (Figs. 1, S1 and S2) were used in the revised version. The clusterings and phylogenetic positions of the lineages resolved in the new trees are the same with those in the last version, except a slight difference in the relative positions of lineages Mantou 2 and Mantou 3. In the last version, the two lineages were clustered together, while in the revised version, Mantou 2 was separated from Mantou 3 but they remained closely related. The revision in the method of phylogenetic analysis has been added in the Supplementary Information (SI Lines 174-181).

5- Divergence time estimations. The phylogenetic strategy that is applied (with which I don't agree, see comment above), probably yields a correct topology, but definitely has a strong effect on the branch lengths estimation, which in turn has huge implications for divergence time estimates. Therefore, these calculations might not be correct. Also, the authors provide very little detail about which method was applied to calculate the divergence time, which molecular clock was used and why the calibration point for the split *S.cerevisiae* –*S.paradoxus* was set from an interval of 10 to 20 Mya (that is already very rough and based only on mutation rate) to an interval of 4 to 10 Mya. Also, it is not clear which mutation rate and generation time the authors considered, and whether they applied different parameters to domesticated vs wild populations. Or were the domestic populations completely discarded for the full analysis? Please give more details.

>> We thank this review comment, but we did not estimate the divergence times based on the whole genome SNPs and we did not set calibration point for the split *S. cerevisiae* –*S. paradoxus* from an interval of 10 to 20 Mya to an interval of 4 to 10 Mya. Actually, we used the amino acid sequences of 2,757 single copy protein genes in the estimation of divergence times and we calculated that *S. cerevisiae* and *S. paradoxus* were split 4 to 10 Mya based on the estimation that the genus *Saccharomyces* originated approximately 10 to 20 Mya (Liti et al., 2006; Hittinger, 2013).

It is quite challenging to date the origin of the domestic population or specific domestic lineages using molecular clock models of *S. cerevisiae* which employ the mutation rate and generation time of laboratory strains determined experimentally in rich medium (Fay & Benavides, 2005; Lynch et al., 2008). The mutation rate has been shown to be heavily influenced by different factors which may vary significantly in different environments as discussed in Gallone et al. (2016, Cell). The mutation rates and generation times of yeasts growing in nature are unknown. Our assumption is that wild isolates of *Saccharomyces* species growing in the wild may be more or less similarly affected by environmental factors because of their overlapping in ecology and geography in nature and thus their mutation rates may be more comparable than domestic isolates. We therefore tried to estimate the divergence time of the wild lineages based on sequence divergences and then went one step further to estimate the divergences time of the domestic population from its closest wild relative.

However, the dates estimated are not crucial to the main questions addressed in this study. Considering a comment from Reviewer #3, we have removed this part from the revised version.

6- In Figure S6, phylogenetic tree (a), did the authors use *S. castellii* to root the tree? The placement of *K. lactis* vs *S. castellii* is wrong. *S. castellii* is a post WGD species and *Kluyveromyces* splitted from *Saccharomyces* before WGD. Furthermore, other trees look problematic from this point of view (cfr. placement of *Zygosaccharomyces* vs *Torulospora*). Make sure that you are not using divergent paralogs within each species that might wrongly place the species in the tree. Also make sure to add *Candida glabrata* to the analysis. In addition, it would be useful to include in the methods a section describing how the trees were obtained. Because of the short length of these fragments, it might be difficult to recapitulate the correct genealogy (although the divergence is quite high and this problem should be limited) but at least you should discuss the incorrect placements in the text.

>> Thanks for pointing out this problem and we have revised Figure S6. These trees were used to show the possible sources of the HGT or introgression genes, but not to resolve the correct phylogenetic positions of the reference species because only one gene was used to draw each tree. Nevertheless, we should do our best to include all closely related

species and to reflect the correct phylogenetic relationships among the species compared when possible. We have paid great attention to this problem in the revised version by adding more relevant reference species and rooting the trees by more appropriate out groups. Please note that we were unable to use the same reference species in different gene trees because the HGT or introgression genes were not available in all reference species. We were unable to add *Candida (Nakaseomyces) glabrata* as suggested by Reviewer #1 because we did not find the homologs of the HGT or introgression genes in the genome of this species. Please also note that the phylogenetic positions of some species in some gene trees are still ‘incorrect’ compared with the trees drawn from genome or multiple gene sequences. It is quite common that single gene trees cannot reflect correct phylogeny of species, especially when using HGT or introgression genes.

(ii) Content

1- It would be useful to provide more details on the production process of the traditional Asian fermentations mentioned in the manuscript (including at which step fermentation happens and if and how there is re-use of the fermentation product to inoculate a new batch, at which temperature fermentations are usually performed, what ethanol concentration they reach, etc). Also, which of these are industrial processes, and which are small, traditional practices performed by local farmers? This would be especially relevant information for the strains in the ‘Milk’ lineage. Figure S1 shows that these strains are in a sister lineage of the Wine strains, and thus must have diverged from the Wine clade AFTER they split from eg. the Beer 1 clade. Therefore, it is intuitively more likely that these Milk strains are more commercial/industrial strains that were initially domesticated in Europe and later transferred back to Asia, no? Or how do the authors interpret this phylogeny?

Additional information on the life history of the strains and details on the specific fermentation processes would greatly help to interpret the genetic (and even the phenotypic) data.

>> We have provided more information about the traditional fermentations sampled in this study in the Supplementary Information (SI Lines 29-51), including raw materials, temperatures during the fermentation processes and ethanol concentrations reached at the end of fermentation. Baijiu (Chinese distilled liquors), Huangjiu (rice wine) and Qingkejiu (highland barley wine) are produced in industrial scale using traditional methods, while Mantou (steamed bread) and the fermented dairy products are homemade in countryside or remote areas. No matter industrial or home scale fermentations, fermented materials from the last patch are usually used in the new patch as starters. We also cited new references for more details of the fermentations.

As for the Milk lineage, the isolates are not commercial or industrial strains, but are all from local families in remote pastoral areas covering western and northern China and Mongolia. The genetic diversity of the Milk lineage ($\pi = 3.86e-03$, 15 isolates) is much higher than (more than three times) that of the Wine/European lineage ($\pi = 1.04e-03$, 9 strains in Liti et al. 2009; $\pi = 1.12e-03$, 19 isolates in Almeida et al. 2015; and $\pi = 1.59e-03$, 24 isolates in Gallone et al. 2016) as shown in Figures 1 and S1 and Table S3. The data suggest that the Milk lineage is native in Asia. The phylogeny suggests that the Milk and Wine lineages originated from a common ancestor, thus the latter should originate from Asia. Furthermore, the Wine lineage also contains four isolates from grape and orchard soil sampled from western China (and another laboratory in China has isolated more isolates belonging to the Wine lineage from local wineries in western China, personal communications). Our data suggest that it is more reasonable to infer that the European wine strains were possibly transferred from China or Asia, being consistent with the main conclusion of this study. We have discussed this in the revised version (Lines 392-401).

2- As many people are not familiar with Chinese geography, it would be useful to provide as a supplemental figure a map with the geographical origin of each strain. The presence of geographical substructure might have a strong effect on results regarding the diversification of domesticated vs wild strains, or diversification of the different wild lineages. E.g. in the PCA plot (Fig S3), domesticated strains might not separate into different subpopulations as observed for wild strains, because they have been sampled from a smaller area? Moreover, this information could serve as a tool to hypothesize about the initial region where strains were first domesticated?

>> We have provided a supplementary figure (Figure S4) for the geographic distribution of the isolates employed in this study. The figure shows that the geographic distribution of the domestic isolates is wider than that of the wild isolates. This has been mentioned in the main body of the revised version (Lines 152-153). Apparently, the much lower genetic diversity (Table S3) and closer PCA plotting (Figure S6) of the domestic isolates than the wild isolates are not caused by smaller sampling area of the former.

We do not think we can hypothesize the initial region where strains were first domesticated based on our data, given that the domestic isolates were from almost all over China and we are unable to recognize a diversity center in the country for the domestic population. But we think the data support our hypothesis that the domestic population might originate through a bottleneck in China/Asia.

3- Throughout the manuscript it's difficult to interpret the major conclusions regarding the origin of domesticated lineages. It would be interesting if the authors would discuss the relationship between the newly identified domesticated lineages in this study and the domesticated lineages identified in previous studies (Liti et al. 2009, Gallone et al., 2016,

Goncavales et al., 2016) in more detail. Are the authors implying that all the domesticated lineages originated from a common DOMESTICATED ancestor in Asia that was subsequently spread in Europe and further diversified in the e.g. Wine/Beer lineages? Or rather that WILD *cerevisiae* strains travelled from Asia to Europe, and were domesticated there? Especially the placement of the Milk strains in Figure S1 is an interesting observation in this regard. The first option would go against the hypotheses of Almeida et al. 2015, who identified a wild stock of yeast that they believe are the wild genetic stock of domesticated wine yeasts. The second option also implies that *S. cerevisiae* has been domesticated multiple times in different geographical locations (Europe/Asia), which is maybe more intuitive? Nevertheless, this should be stated more clearly and discussed in more detail.

>> We understand this concern regarding our conclusion of the origin of domesticated lineages from a single ancestor and we agree that we need to explain and discuss this conclusion in more detail. In order to accommodate this comment and comment 3 from Reviewer #3, we have added one more unrooted ML tree (Figure S2) from a dataset covering 628 isolates to show the position of the Mediterranean oak (MO) lineage and the relationships between the two major domestic groups (solid- and liquid-state fermentation) and between the domestic and wild populations. We use two trees (Figures S1 and S2) to show the relationships between the newly identified domesticated lineages in this study and the domesticated lineages identified in previous studies (Liti et al. 2009, Gallone et al., 2016, Goncavales et al., 2016). We use a long paragraph to describe their phylogenetic relationships in text (Lines 109-126) and four paragraphs to discuss and explain our major conclusions regarding to origin of domesticated lineages in the revised version (Lines 344-401). We hope we have accommodated all concerns from Reviewers #1 and #3 about the origin of domestic lineages.

All the trees constructed in this study (Figure 1 for the 266 Chinese isolates, Figures S1 and S2 for the 554 and 628 isolates respectively covering almost all sequenced *S. cerevisiae* isolates worldwide) show that the domestic isolates formed a monophyletic domestic population which is clearly separated from the wild population, suggesting the possibility of a single origin of the domestic lineages. We have provided three more other evidences supporting this possibility as discussed in Lines 360-375.

Intuitively, we are unable to infer from these trees that *S. cerevisiae* has been domesticated multiple times in different geographical locations (Europe/Asia), because separate closest wild relatives of different domestic lineages in different locations have not been identified so far. Though the MO lineage is closely related with the Wine lineage and Almeida et al. (2015) believe the former is the wild stock of the latter, the possibility that the European Wine isolates were transferred from Asia cannot be excluded. First, if the MO lineage is the wild genetic stock of the Wine lineage, the

nucleotide diversity of former should be significantly higher than that of the latter. However, Almeida et al. (2015) showed that the genetic diversity of the MO lineage ($\pi = 0.99\text{E-}03$, $n = 31$) is lower than that of the Wine lineage ($\pi = 1.12\text{E-}03$, $n = 19$). Second, the Wine lineage contains four Chinese isolates from fruit and orchards (and another laboratory in China has isolated more isolates belonging to the Wine lineage from local old wineries in western China, personal communications). Third, European Wine isolates share HGT genes with Chinese isolates. Among the three horizontally transmitted regions (A, B and C) which are regarded as domestication fingerprints of wine isolates, one (region B) is present in the wild lineage CHN-VIII and a partial fragment of region C is present in three Chinese isolates in the Wine lineage (Figure 4, Table S8). These HGT genes are absent from the MO lineage (Almeida et al., 2015).

4- Line 224-231. The MK test. How many isolates of *S. paradoxus* have been used in the MK test? Also, can the authors expand the discussion regarding their MK test? It would be interesting to discuss the higher number of genes in purifying selection detected in the domesticated versus wild populations in more detail. How can this signal be linked to differences in population dynamics (expansion vs bottleneck), mating behaviour (sex vs no-sex) and heterozygosity? The way this result has been added to the text so far doesn't have any additional value. More information on the history of these traditional fermentation processes (like mentioned earlier) might help the interpretation of such results. For instance, are these strains continuously growing in rich medium, fluctuating environments, is it season-bound, is there back-slopping?

>> We used one *S. paradoxus* isolate (NRRL-Y17217) in the MK test in the last version, as in Branco et al. (2015) and Liti et al. (2009) which also used one isolate as out group in the MK test. To avoid possible bias, we have redone the analysis using six *S. paradoxus* isolates (NRRL-Y17217, UWOPS 919171, UFRJ 50816, YPS 138, N44, and CBS 432) representing the four lineages of the species. The new data (Table S7) obtained were similar (for purifying selection) with or only slightly different (for positive selection) from those of the last version. Our new analysis confirms that positive selection genes in the wild and domestic isolates were very rare (only 2-3 genes out of > 4,210 genes tested), but the fraction of genes subjected to purifying selection in the domestic population (51.5%) is much higher than that in the wild population (8.9%). We used these data in the manuscript mainly to support our observation that the domestic population of *S. cerevisiae* is probably an outcome of adaptive evolution, while the wild population of the species in nature might be primarily shaped by neutral genetic and genomic drifts.

We performed further GO enrichment analysis on the genes subjected to purifying selection in the wild and domestic isolates (Table S7b). In addition to the significant difference in the fraction of genes subjected to purifying selection between the wild and domestic isolates, we found that the genes in purifying selection usually belong to

different specific GO classes in the two groups. Notably, the genes associated with chromosome separation are enriched in the genes subjected to purifying selection in the wild isolates but are not selected in the domestic isolates, probably due to the reduced sexuality in the domestic isolates as shown in the manuscript. A considerably higher number of genes subjected to purifying selection in the domestic isolates are associated with regulation of different biological processes (Table S7b), suggesting stronger constraints in the gene regulation level to the domestic isolates living in nutrient rich environments.

We have revised the table summarizing the MK test results (Table S7) and added a few lines to discuss the new results (Lines 251-258).

5- Figure 4. Introgression analysis. The database to which the fragments are blasted is very limited and the results are therefore misleading. E.g. fragment 1 in table S6: this fragment is present in CHN-IX lineage with 100% identity percentage across strains. When you blast it against your custom database your top hit for 2/3 of the length of the fragment is *K. lactis*. But, it matches for 71%. Therefore, this fragment is obviously not a fragment from *K. lactis* and reporting it as top match is very misleading even if you report it as closely related to *K. lactis*. The authors should vastly extend their custom database and look for better top matches or you have to report this fragment as unknown source. In addition, in Figure 4 legend (top right) the percentage cut-offs are confusing and sound a bit arbitrary (e.g. *K. lactis* is reported only once – pink line on the top of the graph – for a hit that in the table has 84% identity, while in the legend it states ‘*K. lactis* < 90%’).

>> This is a misunderstanding probably due to our unclear method description. We used our custom database to screen possible HGT or introgression events in the first step and then confirm the candidates by BLAST search through the NCBI database and phylogenetic analyses as shown in Figure S11.

Here we use Fragment 1 as an example to show the process and to answer this review comment. The screenshot of the BLAST result of Fragment 1 through GenBank with all the six hits is shown below. The top matches are three *S. cerevisiae* isolates sequenced in Strobe et al. (2015) (YJM1388 was from fermented tapioca in Malaya and YJM1389 and YJM1592 were from sewage in Thailand, and they were all clustered in the Sake lineage). Below the three *S. cerevisiae* sequences are two sequences from *K. lactis* with 71% identities. Within our 266 isolates sequenced, this fragment occurs only in the CHN-IX lineage and in one domestic isolate (JZ10.1 in the Mantou 7 lineage, Table S8, and revised Fig. 4) with 100% identity. The result clearly shows that this fragment represents a HGT event from a source closely related to *K. lactis* among the organisms documented so far in GenBank. The sequence identity (71%) clearly shows that Fragment 1 is not from

K. lactis, but it is the closest species (top match) to the donor of this fragment in GenBank documented so far.

The possible sources of the other HGT and introgression fragments were determined in the same way as for Fragment 1. Please note that we only briefly mentioned and discussed *S. cerevisiae* strains such as YJM1388, YJM1389 and YJM1592 sequenced in other studies that share the HGT or introgression fragments with our strains in the last paragraph under the subtitle Introgression and horizontal gene transfer (HGT) analyses in the revised Supplementary Information (SI Lines 460-469). We think a systematic survey of HGT or introgression events in all the sequenced *S. cerevisiae* strains with different ecological and geographic origins is worthy of a separate study.

We have revised Figure 4 to show the distribution of the HGT and introgression fragments with top matches from species other than *S. cerevisiae* to make sure that they are from alien sources. We also have reduced the complexity of the figure, leaving detailed information including fragment lengths and exact sequence identities in the 1-kb mapping window in Table S8.

BLAST Results

Edit and Resubmit Save Search Strategies ▶ Formatting options ▶ Download YouTube How to read this page Blast report des

Job title: Frag.1

RID 40R3SMMD014 (Expires on 12-26 09:03 am)

Query ID lc|Query_122015 **Database Name** nr
Description Frag.1 **Description** Nucleotide collection (nt)
Molecule type nucleic acid **Program** BLASTN 2.7.1+ ▶ Citation
Query Length 846

Other reports: ▶ Search Summary [Taxonomy reports] [Distance tree of results] [MSA viewer]

+ Graphic Summary
- Descriptions

Sequences producing significant alignments:
 Select: All None Selected:0

↑ ↓ Alignments Download GenBank Graphics Distance tree of results

	Description	Max score	Total score	Query cover	E value	Ident	Accession
[ ]	Saccharomyces cerevisiae YJM1592 chromosome I genomic sequence	1526	1526	100%	0.0	100%	CP004500.1
[ ]	Saccharomyces cerevisiae YJM1388 chromosome XI sequence	1521	1521	100%	0.0	99%	CP005354.2
[ ]	Saccharomyces cerevisiae YJM1389 chromosome I genomic sequence	1521	1521	100%	0.0	99%	CP004474.2
[ ]	Kluyveromyces lactis uncharacterized protein (KLLA0_C19547g), partial mRNA	361	361	90%	1e-95	71%	XM_453073.1
[ ]	Kluyveromyces lactis strain NRRL Y-1140 chromosome C complete sequence	361	361	90%	1e-95	71%	CR382123.1
[ ]	Nonomuraea sp. ATCC 55076, complete genome	55.4	55.4	14%	0.003	70%	CP017717.1

The results of BLAST search through GenBank using Fragment 1 as the query, optimizing for ‘more dissimilar sequences (discontiguous megablast)’.

6- Line 203-210 and discussion. The discussion regarding FLO genes and the correlation to the flocculation phenotype for the strains analysed, sounds a bit out of place because

flocculation has not been assessed. Moreover, while the authors argue that flocculation is not important in industrial strains, it is of huge importance in eg. beer brewing, bioethanol production and winemaking (eg. sherry). In addition, the phenotypic variability observed for this phenotype is not necessarily linked to changes in copy number, but it is known that there are huge differences in activity of the different gene variants and their expression. In addition, it should be noted that the FLO genes are not only responsible for flocculation, but also are involved in pseudohyphal growth, a potential survival strategy for wild yeasts (see the paper by Magwene et al where they hypothesize that sporulation and pseudohyphal growth are two survival strategies for wild yeast). It would be interesting if the authors would measure flocculation and pseudohyphal growth to strengthen the conclusions of their genetic analysis, and possibly even hypothesize on the role of these traits in wild/domesticated strains.

>> It is true that we did not assess flocculation of our isolates and we agree that it will be very interesting to do this experiment, but we think our discussion regarding the result of CNV analysis of FLO genes is relevant due to the following considerations: 1) FLO genes have been well studied in yeast and readers will be interested in them; 2) the clear CNV difference of FLO genes between the domestic and wild populations has not been reported before; and 3) this novel finding seems to be contrary to current industrial practice, thus we should provide explanations. We think our explanations are reasonable regarding the living environments of the wild and domestic yeast isolates. The wild isolates in nature live in more rigorous and fluctuant environments in terms of temperature, nutrition and water availability than domestic isolates which live in nutrient rich and relative stable environments. The cell adherence and biofilm formation are more important for yeast to survive in the wild and thus FLO genes have been maintained in the wild isolates. In contrast, planktonic cells may have advantage for their proliferation in fermentation environments, especially in spontaneous fermentation processes with other microbes and in solid-state fermentation processes. The FLO genes are therefore redundant for domestic isolates and are deleted. The enhanced flocculation ability of some industrial isolates for beer, wine and bio-ethanol production is apparently a domesticated trait due to strong artificial selection for facilitating yeast cell separation after liquid-state fermentation. These fermentation processes usually use elaborately bred pure yeast cultures. In solid-state fermentation (most of our domestic isolates belong to this group) and in liquid-state fermentation processes without yeast cell separation step (e.g., dairy product fermentation), flocculation is apparently not a required trait and thus FLO genes are redundant. We have added a few lines to the Discussion part (Lines 473-486) in the revised version to explain this.

The CNV patterns of FLO genes observed in this study also seems to be inconsistent with the observation and hypothesis of Magwene et al. (2011, PNAS), which showed a negative relationship between sporulation efficiency and pseudohyphal development.

They showed that wild isolates with low heterozygosity and high sporulation efficiency belonged to the pseudohyphae-negative group, while domestic (clinical) isolates with high heterozygosity and low sporulation efficiency belonged to the pseudohyphae-positive group. Since FLO genes are generally considered as being able to promote adhesion and pseudohyphal growth in yeast (Halme et al., 2004, Cell; and many other studies), contrary to Magwene et al. (2011), our result predicts that wild isolates (with FLO genes maintained) will have stronger adhesion and pseudohyphal growth ability than domestic isolates (with FLO genes contracted).

[Redacted]

7- Line 332-336. The single origin hypothesis is not really supported, especially not by point 1 and 2 (or at least not by the limited information and discussion that is given for these points). For point 3, common expansion and contraction of certain genes, e.g. maltose, might indicate the presence of one maltose adapted ancestor but does not exclude the presence of multiple ancestors that adapted to maltose, especially if this statement is based on the CNV of only one single maltose gene Mal31, present in highly variable regions (subtelomeres).

>> We propose the single origin hypothesis mainly based on the phylogenetic trees (Figs. 1, S1 and S2) from genome wide SNPs which show the close relationship and monophyletic nature of all the major domestic lineages recognized worldwide so far and their clear separation of the wild population. Please see our response to Comment 3 above. Here we provide three more evidences supporting or being consistent with the hypothesis. For point 1, we have done a demographic analysis to support the bottleneck (Fig. S7). For point 2, we think the striking difference between the domestic and wild lineages in the degree of heterozygosity is consistent with the hypothesis, and based on this point the possibility that the ancestor of the domestic lineages was formed by outcrossing between different wild lineage, as already hypothesized by Magwene et al. (2011 and 2014), is discussed subsequently (Lines 412-414). For point 3, we listed not only MAL31, but also other MAL genes and genes associated with other networks as shown in Figs. 3 and S10 and Table S6. We use MAL genes here as examples and are unable to mention all genes. We believe the sharing of expanded MAL genes and positive and elevated maltose utilization ability of the Milk and Wine lineages (maltose is absent in their living niches) with the other domestic lineages (maltose is dominant in their living niches) is a strong evidence supporting the common origin of the domestic lineages.

However, considering the comments from both reviewers, the propose of the single origin hypothesis is probably too strong and was not shown straightforwardly in the trees (some fruit and clinical isolates are located between the wild and domestic populations), we

have changed the title by deleting the words ‘single ancestor’ and revised the hypothesis as that the domestic lineages documented so far probably originate from one or two ancestors. The trees clearly show that the domestic lineages form two distinct and monophyletic major groups (solid- and liquid-state fermentation) and each of them stems directly from a common ancestor. This means that the domestic lineages recognized worldwide so far can be traced back to no more than two ancestors.

8- Line 359-368. What’s the number of private/shared polymorphisms within each domestic lineage? If the author’s hypothesis regarding one single common ancestor derived from an outcross is true, many SNPs/heterozygous sites will be shared across the Asian (and European) domesticated lineages.

>> Thanks to this comment, we analyzed the shared and private polymorphisms between different populations and lineages. The percentage of the shared polymorphisms (33.5%) between the two major domestic groups (solid- and liquid-state fermentation) is significantly higher than those (18.6% and 15.9%, respectively) of the domestic groups with the wild population and than that (19.7%) between the whole domestic and wild populations ($P < 2.2e-16$). The shared polymorphisms (11.7 % on average) between different domestic lineages is also significantly higher than that (1.77 % on average) between different wild lineages ($P = 4.63e-12$). Among the wild lineages, CHN-VI/VII shares remarkably more polymorphisms with the domestic lineages, being consistent with its closer phylogenetic relationship with the domestic population. These data are consistent with the relationships between different populations and lineages depicted in the phylogenetic tree (Fig. 1) and PCA plotting pattern (Fig. S6).

We have added only one sentence to the main text (Lines 161-164) to show the data and summarized the data in a new supplementary Table (Table S4).

9- Line 385-396. The hypothesis on reproductive isolation of wild strains is interesting, but should be supported by experimental data. We suggest to select one representative from each wild lineage and perform crosses to test the hypothesis.

>> We performed extensive intra- and inter-lineage cross tests and electrophoretic karyotype analyses for our wild isolates in our paper published previously (Wang et al., 2012, Mol. Ecol.), which is cited in this manuscript. We found partial reproductive isolation between different lineages. We also showed that isolates with remarkably different karyotypes exhibited reduced hybrid fertility, suggesting the role of chromosomal rearrangements in the establishment of reproductive isolation, being consistent with Liti et al. (2006, Genetics) and Hou et al. (2014, Curr. Biol.).

We did additional crosses between isolates in the newly identified oldest lineage CHN-IX with isolates in lineages CHN-III and CHN-IV, and the results are listed below.

Our previous and present data suggest that the reduced hybrid fertility cannot fully explain the lack of admixture (seemly caused by complete reproductive isolation) in ancient wild lineages and further studies are needed.

We have added a few lines in the paragraph concerned (Lines 447-453) in the revised version to accommodate this comment.

Cross	Spore viability (%)
XXYS1.4 (CHN-IX) x XXY26L.1 (CHN-IX)	61.8
XXYS1.4 (CHN-IX) x HN9 (CHN-III)	13.7
XXY26L.1 (CHN-IX) x HN9 (CHN-III)	20.0
XXYS1.4 (CHN-IX) x BJ15 (CHN-IV)	35.3
XXY26L.1 (CHN-IX) x BJ15 (CHN-IV)	38.0
XXYS1.4 (CHN-IX) x BJ19 (CHN-IV)	36.0
XXY26L.1 (CHN-IX) x BJ19 (CHN-IV)	No fertile ascus found

Minor comments:

Figure 1A/Figure S1. Phylogenetic reconstruction. Bootstrapping values or any other type of confidence values are missing for all the phylogenies reported in the text. Please provide confidence values.

>> All the bootstrap support values for the major dots and identified lineages are 100 %, except for Mantou 7 in Figure 1 (bootstrap = 80 %) and Figure S1 (bootstrap = 87.5 %). The legends to these two figures have been revised to indicate this.

Figure 2. Can you explain the heterozygosity for the haploid strains? Is this a PI staining error or rather sequencing errors?

>> Only three domestic isolates (HN2.2 between lineages Qingkejiu and Mantou5; HQ3.1 in lineage Huangjiu; and GS3.1 between Mantou 3 and 4 in the tree) with relative DNA contents of 1.15-1.40 were identified as haploid isolates but showing signals of heterozygosity. They are probably not real haploidy. The problem was probably due to PI staining or FACS measurement errors and is mentioned in the revised Supplementary Information (SI Lines 367-371).

Figure 3. It's very hard to gain any clear insight from this figure. Why not sorting the genes according to GO categories on the top (annotate which GO category) and have a last column with No GO category? Because some genes will be present in multiple GO categories, maybe you can report only major GO categories?

>> We agree with this comment and have revised Figure 3. In the original figure, we sorted the genes in their CNV patterns from left to right: 1) generally expanded in domestic lineages; 2) expanded in specific (liquid-state fermentation) lineages and deleted in other lineages; 3) contracted in domestic lineages; and 4) contracted only in the liquid-state fermentation group. We think it is useful to show this general pattern and remain the original figure in the supplementary information as Figure S10 in the revised version. We have tried to sort the genes according to GO categories, but we found the readability of the figure was not improved (see the figure below). Thus, in the revised figure we have remained only representative and important genes that are discussed in the text. In the revised figure, individual gene names are provided and we think readers will easily gain a clear insight from this figure. Figure S10 (the original Fig. 3 in the last version) is used to show a general CNV pattern of the 225 genes and Table S6 is used to show the detailed information of the genes. The major GO categories of the genes have been added in Table S6.

Copy-number variation (CNV) of 225 genes among the 266 wild and domestic isolates of *S. cerevisiae* sequenced in this study. The genes are grouped according to their GO catalogues

Figure 4. In general, this figure is difficult to follow and the meaning of each colour code should be clearer. For example, one way to remove a layer of complexity is to sort the fragments based on species assignment rather than the arbitrary numbering on the bottom. Also, is it surprising that some fragments are present in all the strains? Can these really be defined as introgressions?

>> We agree that the original Figure 4 was too complicated. We have reduced the complexity of this figure. For the sake of readability, it is impossible to include all the 79 possible HGT or introgression fragments listed in Table S8, thus, the fragments with top matches to *S. cerevisiae* strains (with < 90% identities) are not included in the revised figure. The exact sequence identities which are available in Table S8 are also removed from this figure. The fragment that was present in all the strains in the original figure was not an introgression but was a 1-kb window within a fragment. This window was at the foremost part of Fragment 39 and has been removed.

Figure S4a. Can you replace the y axis name from heterogeneity to heterozygosity?

>> Done.

Figure S4b. Sporulation should be replaced to “Sporulation efficiency” on the y axis and converted to percentage

>> Done.

Figure S4c. Spore viability on y axis should be converted to percentage

>> Done.

Figure S6: tree a/b *Saccharomyces castellii* is misspelled

>> Thanks for pointing out this mistake. In the revised version this figure is Figure S11. This species is not included in the revised Figure S11a and in Figure S11b its current name *Naumovozyma castellii* is used.

The separation liquid/solid state fermentations is probably a bit misleading. Intuitively, this terminology implies that strains belonging to the two groups were specifically adapted to the two types of processes, while for example Chinese bioethanol strains are present in the solid state fermentation group, and eg. all European bread strains are in the ‘liquid’ group. Would it make more sense to call them ‘Asian’ vs ‘European’ domesticated groups? The only problem with that terminology is the unexpected placement of the Milk lineage in the tree as a sister clade of the Wine lineage (see also other comments).

>> We respectfully disagree with this comment. The separation of the two domestic groups is clear and we should assign a suitable name to each of them based the origins or sources of the majority of the isolates included. We think it is suitable to name them as solid- and liquid-state fermentation groups because the former includes the majority of the isolates associated with solid-state fermentation (Mantou, Baijiu, Qingkejiu, and Huangjiu/Sake) and the latter include the majority of the isolates associated with liquid-state fermentation (Wine, Beer, and Milk). It is not suitable to call the liquid-state fermentation group as ‘European’ group because the majority of the milk fermentation isolates in this group are from China and the Wine lineage also includes Chinese isolates. Furthermore, our primary research on African isolates showed that African isolates associated with local honey wine, palm wine and sorghum beer fermentation were all clustered in the liquid-sate fermentation. Only three bioethanol isolates are located in the solid-state fermentation group, but apparently they were developed from a Huangjiu isolate recently. Though all European bread strains are in the ‘liquid’ group, but probably they are all commercial active dry yeast products as the isolates in the ADY lineage, which are usually produced by molasses (liquid-state) fermentation in industrial scale. We have mentioned this in the revised version (Lines 124-126).

It would be interesting to compare the nucleotide diversity for the different lineages measured in this study to those calculated for the other known domestic lineages and report how they compare.

>> Previous studies have recognized five main domestic lineages that contain the majority of worldwide industrial isolates of *S. cerevisiae*, namely Wine, Sake, Beer 1, Beer 2, and a Mixed lineage containing European bread isolates (Liti et al., 2009; Gallone et al., 2015; Gonçalves et al., 2016). The nucleotide diversities of these lineages provided in previous studies are summarized in a table below. In addition, the Mediterranean oak lineage is also included. Only the Wine lineage shared by all studies is comparable in terms of nucleotide diversity. The nucleotide diversity (**1.18E-03**) of the Wine lineage containing only **four** Chinese isolates in this study is similar to those in Liti et al., 2009 (**1.04E-03**, **nine** strains) and Almeida et al., 2015 (**1.12E-03**, **19** strains) but slightly smaller than that in Gallone et al., 2016 (**1.59E-03**, **24** strains). However, it is not clear how the nucleotide diversity of the Wine lineage in Gallone et al. (2016) was calculated because 30 strains including **20** wine and 10 other (beer, spirit, sake and clinical) strains were included in the Wine lineage in their phylogenetic tree. The other five wine strains sequenced in Gallone et al. (2016) were located in other lineages. Therefore, we think the comparison of the nucleotide diversity for the different lineages measured in this study with those calculated for the other known domestic lineages is not so relevant.

Lineage	No. of strains	$\pi \times 100$	$\theta_w \times 100$	References
Wine	9	0.104	0.111	Liti et al., 2009

Wine	24	0.159	0.215	Gallone et al., 2016
Beer 1				
Britain	26	0.313	0.188	
United States	10	0.231	0.172	
Belgium/Germany	18	0.312	0.219	
Beer 2	21	0.295	0.277	
Mixed	17	0.435	0.269	
Wine	19	0.112	0.145	Almeida et al., 2015
Mediterranean oak	31	0.099	0.125	

Line 176. Ploidy measurements. Can you clarify in which way did you combine the ploidy measurements obtained from PI staining together with the sequencing coverage?

>> We determined the relative DNA content of an isolate based on PI staining and estimated the basal ploidy. We estimated chromosomal copy number variation (CNV) based on corrected and normalized sequencing coverage (actual CNV value, V_a). If we did not detect any chromosomal CNV and the relative DNA content did not deviate from 2.0 (or 1.0, 3.0, 4.0) significantly, the isolate was judged as euploid $2n$ (or $1n$, $3n$ or $4n$). Because of the inaccuracy nature of PI staining, it is not sensitive enough to detect the duplication or deletion events of individual chromosomes. If we clearly detected duplication or deletion of certain chromosome(s) from sequencing coverage data, we regarded the isolate as aneuploidy with related chromosome(s) duplicated or deleted, even though the relative DNA content did not correspondingly deviate from euploidy level significantly. Because of our high sequence coverage (average = 193x; median = 190x), we mainly relied on sequencing coverage data for the judgement of individual chromosome duplication or deletion events. Actually, we performed a statistic dispersion analysis of the actual CNV value (V_a) of every chromosome in all the 266 isolates compared, and estimated the copy numbers of individual chromosomes in a specific isolate according to the criteria calculated from an equation as described in the Supplementary Information (SI Lines 352-365).

Line 273 – (The majority of these phenotypic variations are also clearly correlated with specific genomic variations). I think this statement is not sufficiently supported by data (only 3 examples; galactose, melibiose and maltose).

>> Not only 3 examples, raffinose is also included in Figure 5. Because of the length limitation, we are unable to describe the details of all the results we obtained. We present and discuss the maltose and galactose utilization and corresponding genomic variations in details in the main text and move those for melibiose, raffinose, sucrose, together with the results of the tolerance to high temperatures (40C and 41C) and ethanol to the

Supplementary Information (SI Lines 497-534) as mentioned in the main text (Lines 327-329).

Line 291: Include statistics.

>> Done.

Line 299-300. The authors do not show any specific genome or gene changes associated with raffinose and sucrose utilization, tolerance to high temperature and 9% ethanol. This statement is not supported by data.

>> As mentioned in a response above, raffinose and related gene changes are included in Figure 5. Because of length limitation, we show the specific genome or gene changes associated with sucrose utilization and discuss the melibiose and raffinose utilization and related genomic changes in detail in the Supplementary Information (SI Lines 496-533). The tolerance to high temperature and 9% ethanol associated with specific lineages or environments is shown in Figure 5 and discussed in the Supplementary Information (SI Lines 531-533). The correlation of the tolerance to high temperature and 9% ethanol with specific genome or gene changes is not so clear, thus, for more accuracy, we have revised the sentences related (Lines 328-329).

Line 303. The authors should mention the arguments supporting the Gondwanan origin hypothesis and how their conclusions rather support a Far East Asian origin.

>> The Gondwanan origin hypothesis was proposed based on the studies on *S. uvarum* and *S. eubayanus* without Asian isolates. Since our study is only on *S. cerevisiae*, it is not relevant to discuss the origin of the genus and other *Saccharomyces* species. Therefore, the studies on the origin of other species have been removed.

Line 330-331 – (...implying the possibility that the domestic lineages recognized so far originate from a single isolate that originally existed or formed in such man-made environments.). In this statement do the authors refer to the domesticated populations in China or all the domesticated populations recognized so far for *S. cerevisiae*?

>> Please see our detailed responses to the comments on the single origin hypothesis from Reviewers #1 and #3. We refer to all the domesticated lineages recognized so far for *S. cerevisiae*.

Line 370 – (... advantage of heterosis for living in nutrient rich fermentation environments?). Can the authors elaborate this statement or give a citation?

>> We have added a citation Plech et al. (2014, G3) who showed that heterosis is prevalent among domesticated but not wild strains of *S. cerevisiae*.

The authors should check the numbering of supplementary tables in the text and in the figure legends. There is some mixing between table S6 and S7: two tables are labelled S6 (MK test and introgression)

>> Thanks for pointing out these errors. We have checked the numbering of the tables and figures in the main body and SI carefully and hope these errors will not occur in the revised version.

Reviewer #2 (Remarks to the Author):

In this paper, the authors sequence 266 *Saccharomyces cerevisiae* isolates from diverse sources in China. These strains are mixed between domesticated and wild strains. Using their data, as well as comparison of their data to other recent resequencing datasets from *S. cerevisiae*, they obtain new insights into the origin and evolution of this important model organism that also play a key role in human food and drink production.

I enjoyed reading this paper. In my opinion, the methods were appropriate, the results were novel, and the conclusions were in line with the results.

>> Thanks for this generally positive comments.

Regarding my specific comments:

1. Although the text is clear, the wording could be tightened. As an example, in the summary alone, the first sentence probably doesn't need the word 'as' at the beginning, the second sentence says 'origin' twice, the third sentence is missing the word 'on' near its beginning, and the fourth sentence says 'probably' twice. This type of critical editing could be extended throughout the manuscript.

>> Thanks for pointing out the language problem. We have checked and revised the wording carefully across the whole manuscript and will invite by a native English speaker to copy edit the final version when it is acceptable.

2. I felt the authors could get more succinctly consolidate past work on yeast population genomics and more quickly and effectively make their important point. Uncertainty regarding the origins of different yeast lineages and the role of selection vs drift in ongoing evolution in this species reflect a sample issue. Specifically, China had not been sampled extensively and it turns out China is arguably the most important place for unraveling all these issues. The authors seem to try to imply this in the paragraph from lines 57 to 69, but they need to explicitly make their point. The authors need to just directly spell things out for the many readers who will have no familiarity with their research area.

>> We have slightly revised the second and third paragraphs and combined them into one for succinctly consolidating past work on yeast population genomics and to point out the major problems remain to be addressed. In the following paragraph we point out the main reasons why these problems were not resolved in previous studies and why we can address these problems in this study. Then in the last paragraph of the Introduction, we say how we address the problems and the major points we want to address.

3. In the section on gene expansion, contraction, etc., I am not sure why the FLO genes are classified as involved in stress response. These genes tend to be involved in multicellular phenotypes, which can, but don't have to, be induced by stress.

>> Yes, FLO genes are usually regarded as being involved in multicellular phenotypes, but the flocculation of yeast cells is usually a response to environmental changes or stresses, including nutrient starvation and ethanol toxicity (Soares, 2010). Yeast flocculation is usually considered as an important mechanism of stress resistance and a mechanism of protection to harmful environments (Hope et al, 2017). Here we use a broad classification for genes and we think it is probably suitable to classify FLO genes as involved in stress response.

4. Genes with unknown function. This term is used in the main text, but I wondered if this referred to lack of annotated function on SGD or something else. The reason I ask is Augustus was used for de novo annotation and then genes were BLASTed against different databases of known genes. It wasn't clear to me how many new genes were found relative to the reference genome or other existing *S. cerevisiae* genomes using these de novo annotations? Also, it wasn't clear how often these new genes had ascribed functions? Many genes with unknown functions exist in SGD, but this might not be what the authors meant. Probably the methods and supplement need to be built out a little along these lines, while in the main text it might better to say something more specific than 'genes with unknown functions.'

>> We use the phrase 'genes with unknown functions' two times under the subtitle 'Gene expansion, contraction and purifying selection in the domestic population'. Thanks to this comment, we have provided a short definition of this phrase in the Supplementary Information (SI Lines 115-117). This study is only on *S. cerevisiae* and thus we use the SGD database to determine the function of a gene of a *S. cerevisiae* isolate. When a gene is described as 'putative protein of unknown function' or 'dubious open reading frame' in the SGD database, it is regarded as 'a gene with unknown function' in this study. Please see Table S5 (sheet b) for the descriptions of the genes showing significant CNV among different lineages of *S. cerevisiae* sequenced in this study.

These genes are not new genes. They have been identified as genes or ORFs, but their functions are unknown or uncertain at present. We have found many new genes from our

isolates, but we do not want to include them in this manuscript because they are not directly related with the main topics of this study (diversity, origin and evolution of the domestic population of yeast) and this manuscript is already too long.

5. In Figure 3, it is a bit confusing that copy number is provided as a continuous range from 0 to greater than 3. I realize that noisy coverage data are used to infer copy number but is there some way to obtain a maximum likelihood estimate of the coverage of a given gene in a given strain. It is also a bit hard to look at because the coloring scheme employed has 8-ish levels and my eyes are drawn to the grey, which is the baseline coverage of 1. Anyways, I think the aesthetics of this figure could be improved to better emphasize the result.

>> We agree that the original Figure 3 is too complicated and have revised this figure substantially. In the revised figure we have remained only representative and important genes that are discussed in the text. In the revised figure, individual gene names are provided and we think readers will easily gain a clear insight from this figure. The CNV patterns of the 225 genes and more information are available in Table S6.

We also have reduced the coloring scheme to 5-ish levels according to the criteria based on statistical analysis of the distribution of the CNV value of each gene throughout the 254 isolates compared, as described in the revised SI (SI Lines 256-262) .

6. In Table S6, the header ‘neutral selection’ is used. Obviously there is no such thing, so this needs to be modified. I wasn’t sure what was meant by this, so I didn’t suggest a replacement.

>> We listed the genes that do not subject to any selection (they are neutral) under this header as detected by the MK test. We agree that it is not correct to use the term ‘neutral selection’ here. The aim of the MK test is to check the ratio of genes subjected to selection, thus it is not necessary to list the neutral genes in detail. We have deleted these genes from the revised version of this table (Table S7 in the revised version).

Reviewer #3 (Remarks to the Author):

This manuscript presents an analysis of genomes and phenotype data for new strains of *S. cerevisiae* to answer questions about domestication. The strains are from sources not previously examined (rice wine, steamed bread, distilled liquors – all from china) and provide significant and new insights into the origin and domestication of yeast strains. Overall the manuscript is well presented, thorough in the analysis and represents a large body of work and a strong advance. However, there are a number of key points that should be addressed. I think most of these can be addressed by more careful and conservative statements of the claims made but also inclusion of alternative models in the discussion.

>> Thanks for this generally positive comments.

1. Phylogenetic analysis is not appropriate for individuals in a population. Although prior studies continue to build trees and talk about distinct lineages, it is neither correct nor justified. *S. cerevisiae* strains mate and recombine with one another. We don't talk about humans as forming distinct lineages but rather distinct populations. That said it is acknowledged that trees can be useful ways of presenting data. However, the nomenclature and particularly the use of language like distinct lineage is misleading. These are populations and we know they are interbreeding.

>> We understand this comment regarding the usage of the term 'lineage'. We follow the usage of previous studies on population genetics and genomics of *S. cerevisiae* (e.g., Fay & Benavides, 2005; Liti et al., 2009; Magwene et al., 2011; Almeida et al., 2015; Gallone et al., 2016; Gonçalves et al., 2016). The term 'lineage' has been widely and commonly used in the yeast community and we think it is probably better for us to follow the language of the community. In this study, we call the wild and domestic groups as wild and domestic 'populations' and call the sub-groups within each population as lineages.

2. Single recent ancestor of domesticated strains. The proposal that domesticated lineage arose from single isolate is not supported. The observations used to support this do not exclude other possibilities as well as known admixture/introgression events that support the notion of a domesticated population rather than a single isolate. Later statements, "the heterozygosity shared by almost all domestic lineages is unable to be explained by cloning reproduction". This I agree with and shouldn't have been posed in ref 18. However, taking one step away and suggesting that domesticated lineages arose from a single hybrid is similarly ill conceived, not just because the data are consistent with a model of domestication with outcrossing, but because many individual genes often show histories that are not consistent with a single origin.

>> This comment is shared by Reviewer #1. Please see our responses to Comments 3 and 7 for 'Content' from Reviewer #1 above. Here we explain our hypothesis again. The trees constructed in this study based on genome wide SNPs (Figs. 1, S1 and S2) show that all the recognized domestic lineages are closely related and clearly separated from the wild population. The domestic lineages form two distinct major groups (solid- and liquid-state fermentation) and each of them stems directly from a common ancestor. This means that the domestic lineages recognized worldwide so far can be traced back to no more than two recent ancestors. We further propose the single origin hypothesis based on the clear separation of the domestic from the wild populations and additional observations that are summarized in the Discussion (Lines 362-377), including the CNV patterns of many genes that are consistent with a single origin hypothesis.

On the other hand, from our results we are unable to find any evidence supporting the multiple origin hypothesis of domestic lineages of *S. cerevisiae*. Separate closest wild relatives of different domestic lineages in different locations have not been identified so far. The only problem is the Mediterranean oak (MO) lineage which is discussed in the revised version of this manuscript (Lines 378-391) (please also see our response to Comment 3 for Content from Reviewer #1 above).

However, considering the comments from both reviewers, the propose of the single origin hypothesis is probably too strong and was not shown straightforwardly in the trees (some fruit and clinical isolates are located between the wild and domestic populations), we have changed the title by deleting the words 'single ancestor' and revised the hypothesis as that the domestic lineages documented so far probably originate from one or two ancestors.

3. The claim that liquid and solid domesticated yeast strains arose from a single ancestral domestication event is not well supported. However, there is strong evidence that there was a single event in the origin of domesticated strains from China presented in this study. However, this quite likely did not include european/wine strains. First, it doesn't account for wild mediterranean oak populations. Second, not many wine strains were used in the analysis. The observations that the MAL genes are present in the milk and wine strains is certainly interesting and does point to some shared ancestry between the European wine and Asian domesticates, but it does not imply an origin of these strains in China.

>> We have included the majority of non-Chinese domestic *S. cerevisiae* strains including more than 40 wine strains that have been sequenced in previous studies (Liti et al., 2009; Strobe et al., 2015; Gallone et al., 2016; Gonçalves et al., 2016) in the trees (Figs. S1 and S2) covering a total of 554 and 628 strains, respectively. The Mediterranean oak (MO) isolates identified in Almeida et al. (2015) are included in the tree covering 628 strains (Fig. S2). The problem of the MO lineage and the possibility of the European Wine isolates coming from Asia have been discussed in the revised version (Lines 378-391). Please also see our response to Comment 3 on Content from Reviewer #1 above. The shared ancestry of the European wine lineage with the Milk lineage which is native in Asia and the clustering of Chinese wine isolates together with European wine isolates support the origin of the latter from China.

4. The dates for the divergence time need to explicitly state what assumption is made, I would prefer generations/yr. The reference to Hittinger (2013) for 10-20 Mya is not good, this is a review and is based on calibrations in plants and animals and perhaps one fungal fossil over 100 Mya. Whatever the solution, either remove the dates or make explicit in the text (not supplement) that these numbers are quite questionable.

>> We agree that it is quite challenging to estimate the divergence times and the dates are not crucial to the main questions addressed in this study, thus we have removed this part in the revised version.

5. MK test results are misinterpreted. $NI > 1$ indicates $Pn/Ps > Dn/Ds$. Assuming syn. sites are neutral this means either Pn is inflated or Dn is deflated. Dn can be low due to purifying selection – but this is divergence between species, which is shared by both domestic and wild strains and so wouldn't explain a difference between the two. Pn can be inflated due to loss of selective constraint, i.e. weaker purifying selection. The result of reduced purifying selection in domesticated strains has been found previously in yeast, e.g. Elyashiv et al. 2010.

>> Thanks for point out the possible problems in the interpretation of MK test results. Actually we also considered these problems and did our best to avoid or minimize the possible biases when setting up the gene dataset for MK test by following the requirements (SI Lines 474-482): i) each gene was identified exactly as a single copy gene from assembled genomes; ii) only a gene with $\geq 90\%$ amino acid identity and 80% amino acid coverage thresholds among different lineages was included, the genes showing exceptionally diverged sequences among different lineages or populations were excluded; iii) only strains with genes that meet the requirements above were included, however, each gene set should cover at least 80% of the strains in a given group identified from the phylogenetic analysis to ensure the sampling sizes of each group. Genome wide SNP (923,479 sites) sharing analysis did not found fixed differences between the wild and domestic populations of *S. cerevisiae*, meaning that the divergence between the two populations was not found. According to the arithmetic of Egglib, Dn is the number of non-synonymous fixed differences between ingroup (*S. cerevisiae*) and outgroup (*S. paradoxus*). Ds is the number of synonymous fixed differences between ingroup (*S. cerevisiae*) and outgroup (*S. paradoxus*). Pn is number of non-synonymous polymorphic sites among ingroup (*S. cerevisiae*). Ps is the number of synonymous polymorphic sites among ingroup (*S. cerevisiae*). In order to reduce the bias of MK test further, we used six isolates from the four lineages of *S. paradoxus* representing the maximum genetic diversity of the species as an outgroup. The biases can be further reduced through increasing the diversity and strain number of outgroup.

In the study of Elyashiv et al. (2010), only 16 *S. cerevisiae* strains were compared, including 8 European, 3 Sake, 3 Malaysian, 1 North American and 1 West African strains. The polymorphisms within each lineage was very limited. Since population genetics analyses including the MK test heavily rely on statistical analysis based on large samples, we think that the results obtained from very limited samples with very limited polymorphisms should be interpreted with caution.

6. The beginning of the discussion, there should be a distinction between the Far East Origin of particular species as compared to the domesticated strains. It is fine to discuss both, but the present manuscript only deals with domesticated strains of *S. cerevisiae*. See these suggestions:

Line 302 “origin of some *Saccharomyces* yeasts”

Line 303 “was also proposed recently for other species”

Line 304 “provide evidence” ... origin hypothesis for *S. cerevisiae*”

>> Since this study is only on *S. cerevisiae*, it is not quite relevant to discuss the origin of the genus and other *Saccharomyces* species. Therefore, we have re-written this paragraph and the studies on the origin of other species have been removed.

Other comments

Line 19. “Based on genomic”

>> Done.

Line 58. “could be due to insufficient”

>> Done.

Line 60. “A recent” - the tone is too combative in this paragraph. I think these changes would help.

>> Done. Thanks

Line 66-67. Check Naumov's work, I believe he has made this hypothesis as well.

>> Thanks, two papers Naumov & Nikonenko (1988) and Naumov et al. (2003) have been added.

Line 134. remove “that consistent of”

>> The whole sentence has been re-written (Line 138).

Line 173. The negative correlation – it would be worth citing Magwene (2011) who had a similar result.

>> Done.

Line 208 identified rather than recognized.

>> Done.

Line 283 and other places discussing the Maltose result – Naumov has (I believe) stated this in past work that Maltose is a strong indicator of domestic vs wild yeasts.

>> We have reviewed Naumov's papers available to us and have not found a paper that shows Maltose is a strong indicator of yeast domestication. We have sent an email to Dr. Naumov for this question but have not received his response.

Figure 1 – the structure analysis on the bottom is not useful since one can't tell which strains the bars refer to. The full info is in the supplement so the Fig 1 panel can be removed.

>> We use Figure 1b to show the population structure of *S. cerevisiae* in China. It shows if the lineages identified in the tree are clear or share polymorphisms with other lineages. This information cannot be shown in the tree. The names of the lineages in both parts are the same and it is easy to correlate them with each other. We think it shows important information and remain it in the revised version.

Reviewers' comments:

Reviewer #1 (Remarks to the Author):

We thank the authors for their extensive response to the raised comments, and the effort they did adjusting the manuscript accordingly. I really believe the quality of the manuscript is increased now, but there are a few additional important points that need attention. I've listed these points below, in some cases preceded by the corresponding section from the rebuttal.

1 - The text can still be improved in terms of grammar, writing style, etc.; I would advise an additional check-up by a native speaker

2 - The references to the figures (especially the supplemental figures) are still not correct; please correct.

3 -FigS3: I would add strain names in the supplemental version of the structure plots. In addition, the wording of Figure S3 is a bit unclear, and in the main body text Line 137-148, the numbering of the Figures is missing the name of the panel (a,b,c).

4 - Replace "domestic" with "domesticated" throughout the text. Domestic refers more to something "local".

5 - The authors refer to the Mantou, Baijiu, Huangjiu etc. isolates as domesticated strains very early in the text, whereas they show evidence for the "domestication" later on in the results section. Maybe they should refer to these strains as fermentation-associated strains or Asian-fermentation processes.

6 - Linkage disequilibrium analysis: I am still not convinced about this. Linkage disequilibrium is calculated based on reads mapped to the reference sequence (s288c) that is just a proxy of the real genome structure of the strain. These strains have been sequenced at their natural ploidy and you have a mix between haploid, diploid, tetraploid strains that on top of this complexity will have variable copies of certain chromosomes and variable copies of certain regions within the ploidy context (Figure 2). If you detect a duplication by mapping your reads against the reference sequence, that duplication might be present somewhere else within the genome of the target strain, but you are including those reads in you LD calculations. The same is true for other forms of rearrangements. In short, the number of haplotypes or unphased genotypic allele counts has a big influence on the performance of LD estimators and I do not think that the way you carried on the analysis is correct. At the same time I do not have a solution to deal with this kind of complexity. Probably exclude complex regions from the analysis would be the first step (as it happens for human genomes). But this would not be enough.

The authors state: "We only use the LD analysis to show that recombination is more frequent in the domestic population than in the wild population, being consistent with the structure analysis"

Other confounding effects might be crucial, first of all population substructure: are mosaic strains included in the analysis? Looks like domesticated and wild isolates have been treated in bulk without accounting for subpopulations and admixture. Mosaic strains can introduce an artificial LD signal. Second, can you discuss the meaning of lower linkage disequilibrium decay detected for domesticated strains and your statement that is due to increased recombination rate? (Line 157-158) How is this connected to low sporulation efficiency/low spore viability measured in domesticated strains?

This needs to be discussed further or removed from the analysis.

7 - Can you annotate Figure S7 better (e.g. legend for the color of wild vs domesticated panel (a) , and the axis labels in the other panels should be annotated as reported in the figure legend.

8 - Line 126-127 main text: everything originates from a common ancestor, they are rather 'closely related'. I would rephrase this statement.

9 - The MK test: the much higher number of genes under purifying selection is per se' very interesting but there is still a lack of interpretation of this result. If you have a $NI > 0$ and you have no sign of positive selection (as often observed in *Saccharomyces cerevisiae*, see: Fay JC. Weighing evidence for adaptation at the molecular level. Trends Genet 27: 343-9) it's possible that your P_n is increased due to the presence of slightly deleterious mutations (with low probabilities of fixation) that can come from

a recent bottleneck and consequently a reduction of effective population size as you state in your paper for the domesticated population. Did you also use a cut-off on the frequency of the substitution? Is this value corrected for segregating vs fixed substitution across the two groups? In addition, what can be the effect of "sex" vs "non-sex" and mutation rate between the 2 populations on this analysis? All these aspects should be better discussed in the text because you are trying to support a very crucial point: adaptive vs neutral evolution.

10 – GO enrichment on genes under purifying selection (Line:250-257): the link between the GO enrichment results and the interpretation is not clear, and maybe too strongly put. E.g. you are taking a bunch of "different biological processes" that can mean anything, from regulation of protein phosphorylation to cellular polysaccharide metabolic process, from alpha-amino acid metabolic process to cellular amino acid biosynthetic process (just picking from the top 10 of table S7b), suggesting very negative selection at the WHOLE gene regulation level because of nutrient rich environment in domesticated isolates? Line:250-257. Please discuss this further and better or remove.

11- In the previous revision I raised some concerns regarding the Asian origin of all domesticated lineages of *S. cerevisiae*. My text from the first revision: "throughout the manuscript it's difficult to interpret the major conclusions regarding the origin of domesticated lineages. It would be interesting if the authors would discuss the relationship between the newly identified domesticated lineages in this study and the domesticated lineages identified in previous studies (Liti et al. 2009, Gallone et al., 2016, Goncavales et al., 2016) in more detail. Are the authors implying that all the domesticated lineages originated from a common DOMESTICATED ancestor in Asia that was subsequently spread in Europe and further diversified in the e.g. Wine/Beer lineages? Or rather that WILD *cerevisiae* strains travelled from Asia to Europe, and were domesticated there? Especially the placement of the Milk strains in Figure S1 is an interesting observation in this regard. The first option would go against the hypotheses of Almeida et al. 2015, who identified a wild stock of yeast that they believe are the wild genetic stock of domesticated wine yeasts. The second option also implies that *S. cerevisiae* has been domesticated multiple times in different geographical locations (Europe/Asia), which is maybe more intuitive? Nevertheless, this should be stated more clearly and discussed in more detail"

The new version of the text is more complete but still quite confusing.

To summarize: based on i) the unusual placement of the Milk clade, ii) partial A,B,C introgression present in CHN-VIII iii) higher genetic diversity in the milk clade, the authors conclude indeed that all the domesticated lineages documented so far originate from Far East Asia and that the Wine yeasts were transferred from Asia (and also Beer1 and MO sister lineages of the Wine lineage?). Do the authors imply that the MO clade is not the wild stock of the Wine clade?

One point I do not agree with is the statement that higher nucleotide diversity indicates necessarily an earlier origin. The authors use this argument e.g. to prove that the milk lineage is older than the wine lineage:

'The genetic diversity of the Milk lineage ($\pi = 3.86e-03$, 15 isolates) is much higher than (more than three times) that of the Wine/European lineage ($\pi = 1.04e-03$, 9 strains in Liti et al. 2009; $\pi = 1.12e-03$, 19 isolates in Almeida et al. 2015; and $\pi = 1.59e-03$, 24 isolates in Gallone et al. 2016) as shown in Figures 1 and S1 and Table S3. The data suggest that the Milk lineage is native in Asia.'

Or

'First, if the MO lineage is the wild genetic stock of the Wine lineage, the nucleotide diversity of former should be significantly higher than that of the latter. However, Almeida et al. (2015) showed that the genetic diversity of the MO lineage ($\pi = 0.99E-03$, $n = 31$) is lower than that of the Wine lineage ($\pi = 1.12E-03$, $n = 19$).'

Everything depends on the life history of the yeasts. More generations (e.g. because of serial re-inoculation of the yeasts, such as in beer or milk lineages), automatically results in faster evolution, and thus diversity, compared to yeasts that only undergo a few generations per year (such as the wine yeasts).

Also regarding the HGT. The authors state:

"Third, European Wine isolates share HGT genes with Chinese isolates. Among the three horizontally

transmitted regions (A, B and C) which are regarded as domestication fingerprints of wine isolates, one (region B) is present in the wild lineage CHN-VIII and a partial fragment of region C is present in three Chinese isolates in the Wine lineage (Figure 4, Table S8). These HGT genes are absent from the MO lineage (Almeida et al., 2015)."

"The close relationship of the Mediterranean oak (MO) group with the Wine lineage was also resolved (Fig. S2 and supplementary information). The two sub-groups diverged directly from a common ancestor (Fig. S1, S2)."

It's not really clear to me how this shows an Asian origin. Looking at the close relationship between the Wine, MO, Beer2 clade and Milk, are the authors implying that these HGT regions were lost in the sister lineages of the Wine clade? It's hard to spot the actual relationships between strains from the radial phylogeny in Fig. S2.

The point is that I probably believe in the Asian origin of the domesticated lineages. But the authors fail to explain this hypothesis clearly in the text; they base their assumptions on hints that are very interesting per se' but are put together in a confusing way without considering all possible implications.

Reviewer #2 (Remarks to the Author):

I was satisfied with this revision.

Reviewer #3 (Remarks to the Author):

To the authors,

The manuscript is greatly improved and will be a significant advance in our understand of *S. cerevisiae* diversity. There are, however, some important concerns with the language/interpretation of the results. In some cases I believe a slight change in the wording will provide the necessary accuracy. In other cases it may be a change in the interpretation. It was always clear to me whether this was language or interpretation.

The two main issues are: the 'single or two ancestors' statements and purifying selection in the domesticated group, highlighted by (*) below.

1) I suggest these changes to the abstract for better grammar/clarity:

A previous study showed that wild populationS of *S. cerevisiae* probably originated in Far East Asia. However, the diversity and evolutionary history of domestic populationS of the yeast remain elusive.

we show here that China/Far East Asia is also likely the CENTER OF ORIGIN of domestic populations of the species

The domestic populationS...

2) "The fraction of the ancestral population that entered into the wild and the domestic group is 99.36 % and 0.64 %, respectively."

This isn't clear. Is this the amount of variation that survived the bottleneck. What parameters are specifically estimated? The bottleneck during, change in N?

3) "with an average ratio of heterozygous sites of 0.0055%"

This statement isn't clear. Only in the methods is it mentioned it is the ratio to the genome size. Better to call it the proportion of heterozygous sites in the genome.

4) should be: ", consistent with Magwene et al. (2011)"

5) should be: "only specific lineages; most remarkably,"

6*) The language used to describe MK test is not accurate. The MK test does not determine whether a gene is subject to selection. Indeed, a non-significant MK test is consistent with purifying (negative) selection, which is consistent with a neutral model. The MK tests whether observed polymorphism/divergence is subject to selection. The standard neutral model outlined by Kimura is not rejected by MK test indicating $NI > 0$. $NI < 0$ is not consistent with neutral model.

Prior interpretations of the MK test have been relaxed purifying selection in domesticated lineages leading to more amino acid polymorphism than expected. The way the manuscript is written, it sounds like there is more purifying selection in domestic lineages.

should be: "the remaining (91.0%) were consistent with a neutral model"

The neutral model includes purifying selection.

And in the discussion "purifying selection are rare in the wild population," This is not supported. Purifying selection is stronger in wild than domestic populations as established by prior work and supported by the authors own analysis. See comments on MK test.

7) should be: "C) first found in the wine yeast strain"

8*) Are the population genetic analyses inaccurate due to introgression? I worry that some SNPs calls are for introgressed regions, e.g. from Spar into strain x, yielded artificially high divergence and rates of polymorphism. In particular, CHN-IX, which shows the most extensive introgression with S. paradoxus.

9*) "and each group originates directly from a single recent ancestor (Figs. 1, S1 and S2), implying that the domestic lineages recognized worldwide so far can be traced back to no more than two ancestors."

I'm sorry, but this statement is not supported. When there is no recombination one can trace lineages back to a single common ancestor, e.g. human mtDNA, but this does not imply a bottleneck or anything related to domestication; it is a simple consequence of population genetics and entirely expected. When there is recombination there are multiple ancestors.

I'd suggest changing the wording to .. originating from a single lineage or single population.

That all domesticate yeasts come from either of two lineages/populations is a significant statement – even if very much consistent with most but not all prior work. But to come from one or two ancestors (cells implied) is incorrect or not supported.

"lineages recognized so far originate from a single isolate that" I'd say they have a shared origin (population or lineage implied or explicitly stated), not a single isolate which is unlikely and no data to support or refute.

Similar statements elsewhere would need to be changed.

A general – 'single origin hypothesis' is fine. The supporting MAL and FLO gene analysis provide compelling evidence for a shared origin even if subsequent admixture occurred.

10) typo: "all most all the domestic"

11) should be: "diversity of the former should"

12) Regarding the MO lineage, there are some caveats to the statements made. The higher diversity of wine lineages could be subsequent admixture where MO has had no admixture. The wine-like strains found in china could be migrants from europe. In my view, the last argument is strong, but should be flushed out. The HTG genes in china that define the wine group either formed in europe and then had to have migrated to china, which seems quite unlikely. Or the original HTG occurred in china and the wine strains were derived from this domestic population.

13) Typo "while a considerable NUMBER? of other genes"

14) should be "may have AN advantage"

Responses to Reviewers' comments:

Reviewer #1 (Remarks to the Author):

We thank the authors for their extensive response to the raised comments, and the effort they did adjusting the manuscript accordingly. I really believe the quality of the manuscript is increased now, but there are a few additional important points that need attention. I've listed these points below, in some cases preceded by the corresponding section from the rebuttal.

>> We appreciate the positive comments on our revisions.

1 – The text can still be improved in terms of grammar, writing style, etc.; I would advise an additional check-up by a native speaker

>> The new revised version has been copy-edited by a native English speaker Dr. Timothy James from the University of Michigan, who is an expert on evolutionary biology of fungi.

2 – The references to the figures (especially the supplemental figures) are still not correct; please correct.

>> Thanks for finding these errors. We have double checked the references to the tables and figures to make sure that all of them are correctly labeled and referred in the new revised version.

3 –FigS3: I would add strain names in the supplemental version of the structure plots. In addition, the wording of Figure S3 is a bit unclear, and in the main body text Line 137-148, the numbering of the Figures is missing the name of the panel (a,b,c).

>> The number of strains used in Fig. S3c and S3d are 266 and 628, respectively. The strain names will be too small to read if they are added. To improve the readability and understandability of the figure, we have added lineage names in panels c and d.

We have specified the panels of Figure S3 when cited in the text (Lines 146, 148).

4 - Replace “domestic” with “domesticated” throughout the text. Domestic refers more to something “local”.

>> Done.

5 - The authors refer to the Mantou, Baijiu, Huangjiu etc. isolates as domesticated strains very early in the text, whereas they show evidence for the “domestication” later on in the results section. Maybe they should refer to these strains as

fermentation-associated strains or Asian-fermentation processes.

>> Actually at the very beginning of the Results section (the 1st paragraph) we show the clear separation of the domesticated from the wild isolates. Following this comment, we have used ‘fermentation-associated isolates’, ‘isolates from fermentation environments’ or ‘isolates associated with Mantou, Baijiu, Huangjiu or Qingkejiu fermentation’ in the text when appropriate.

6 – Linkage disequilibrium analysis: I am still not convinced about this. Linkage disequilibrium is calculated based on reads mapped to the reference sequence (s288c) that is just a proxy of the real genome structure of the strain. These strains have been sequenced at their natural ploidy and you have a mix between haploid, diploid, tetraploid strains that on top of this complexity will have variable copies of certain chromosomes and variable copies of certain regions within the ploidy context (Figure 2). If you detect a duplication by mapping your reads against the reference sequence, that duplication might be present somewhere else within the genome of the target strain, but you are including those reads in your LD calculations. The same is true for other forms of rearrangements. In short, the number of haplotypes or unphased genotypic allele counts has a big influence on the performance of LD estimators and I do not think that the way you carried on the analysis is correct. At the same time I do not have a solution to deal with this kind of complexity. Probably exclude complex regions from the analysis would be the first step (as it happens for human genomes). But this would not be enough.

The authors state: “We only use the LD analysis to show that recombination is more frequent in the domestic population than in the wild population, being consistent with the structure analysis”

Other confounding effects might be crucial, first of all population substructure: are mosaic strains included in the analysis? Looks like domesticated and wild isolates have been treated in bulk without accounting for subpopulations and admixture. Mosaic strains can introduce an artificial LD signal. Second, can you discuss the meaning of lower linkage disequilibrium decay detected for domesticated strains and your statement that is due to increased recombination rate? (Line 157-158) How is this connected to low sporulation efficiency/low spore viability measured in domesticated strains?

This needs to be discussed further or removed from the analysis.

>> Thanks for pointing out the problem in LD analysis. We agree that the problem is complicated and it is not easy to find a solution to deal with this kind of complexity.

As we explained in the rebuttal letter for the last version, we used PLINK v1.07 for LD analysis which counts for unphased SNP matrix (Purcell et al., 2007). The result for the wild isolates should be reliable because they are almost all homozygous and thus are equivalent to haploid isolates. The complexity exists mainly in domesticated isolates. The LD decay (with a half maximum at 2.8 kb) of the domesticated group estimated in this study is similar to that (decaying to half its maximum value at 3 kb or less) showed in Liti et al. (2009) who used haploid strains. The majority of the strains compared in Liti et al. (2009) are domesticated ones. Therefore, we think the result of our LD analysis for the domesticated isolates is also reasonable.

The lower LD decay detected in the domesticated strains suggesting more recombination, being consistent with the structure analysis showing more shared polymorphisms and mosaic isolates in the domesticated lineages. We understand that this result seems inconsistent with the low sporulation efficiency/low spore viability observed in domesticated strains. The latter phenomenon implies that the domesticated strains mainly undergo asexual production and thus should have low recombination rate. However, we do have observed more recombination events in structure analyses (Figs. 1b, S1 and S3c,d). Magwene et al. (2011, PNAS) also observed higher rate of crossing and mitotic recombination in their “domesticated” (human-associated) isolates with poor sporulation efficiency than in “natural” isolates.

Considering that we will add a long paragraph to explain and discuss the result of LD analysis for accommodating this review comment but LD analysis is not crucial for the main conclusion and does not contribute significantly to this study, thus we have removed this part in the new revised versions of the main body and SI, as suggested by the reviewer.

7 – Can you annotate Figure S7 better (e.g. legend for the color of wild vs domesticated panel (a) , and the axis labels in the other panels should be annotated as reported in the figure legend.

>> We have revised the legend to this figure (Figure S6 in the new version). We think the meaning of color in panel (a) is straightforward and easy to be understood.

8 - Line 126-127 main text: everything originates from a common ancestor, they are rather ‘closely related’. I would rephrase this statement.

>> We have rephrased the sentences concerned in the new revised version (Lines 133-134) and hope they are clearer now. We used the statement “The two sub-groups of the liquid-state fermentation group diverged directly from a recent common ancestor (Figs. S1 and S2)” (Lines 129-130) to emphasize the monophyletic nature of

the liquid-state fermentation group.

9 – The MK test: the much higher number of genes under purifying selection is per se very interesting but there is still a lack of interpretation of this result. If you have a $NI > 0$ and you have no sign of positive selection (as often observed in *Saccharomyces cerevisiae*, see: Fay JC. Weighing evidence for adaptation at the molecular level. Trends Genet 27: 343-9) it's possible that your P_n is increased due to the presence of slightly deleterious mutations (with low probabilities of fixation) that can come from a recent bottleneck and consequently a reduction of effective population size as you state in your paper for the domesticated population. Did you also use a cut-off on the frequency of the substitution? Is this value corrected for segregating vs fixed substitution across the two groups? In addition, what can be the effect of “sex” vs “non-sex” and mutation rate between the 2 populations on this analysis? All these aspects should be better discussed in the text because you are trying to support a very crucial point: adaptive vs neutral evolution.

>> Thanks for point out the possible bias in the MK test due to the presence of slightly deleterious mutations. But $NI = (P_n/P_s)/(D_n/D_s)$ which will always > 0 . We guess the viewer actually wanted to say the fraction of nonsynonymous substitutions driven by positive selection ($\alpha = 1 - NI$). In contrast to the possibility pointed out by Reviewer #1 that “ P_n is increased due to the presence of slightly deleterious mutations (with low probabilities of fixation) that can come from a recent bottleneck”, Hughes (2007) wrote “that during a population bottleneck, slightly deleterious mutations may no longer be effectively removed by purifying selection, and thus a certain number of such mutations may drift to fixation. As a result, $D_n:D_s$ will exceed $P_n:P_s$ this phenomenon is likely to lead to a high rate of false detection of positive selection by this (the MK) test.” We think the problem of the presence of slightly deleterious mutations is minor in our study because the result of this study is consistent with previous studies (Doniger et al., 2008; Liti et al., 2009; Elyashiv et al., 2010; also see Fay et al., 2011) indicating that positive selection is rarely detected in *S. cerevisiae*.

We understand that one way to cope with the effects of slightly deleterious mutations is to remove low-frequency polymorphisms from the analysis (Fay et al., 2001; Liti et al., 2009; Charlesworth & Eyre-Walker, 2008). However, a variety of different cut-off, from $< 12.5\%$ to $< 20\%$ for humans, *Drosophila* and yeast, have been used, which seem to be quite arbitrary. Charlesworth & Eyre-Walker (2008) investigated the performance of this method theoretically and showed that “although removing low-frequency polymorphisms reduce the bias in the estimate of adaptive evolution, the estimate is always downwardly biased, often to the extent that one

would not be able to detect adaptive evolution, even if it existed”. Therefore, we did not use an arbitrary cut-off on the frequency of polymorphisms.

In order to see if we could use a reasonable cut-off, we performed an additional analysis on the frequency of minor alleles and found that the frequency distribution patterns of minor alleles in the wild and domesticated populations of *S. cerevisiae* are quite different (see the figure below). If an arbitrary cut-off is used, the effect on the wild and the domesticated populations will be different. The minor alleles with frequencies of 0.1 to 0.15 are significantly reduced in the domesticated populations and a skew to minor alleles with a 0.05 frequency is evident in the domesticated populations. This is probably resulted from purifying selection because it is generally believed that purifying selection presents harmful variants from rising in frequency, resulting in a skew in the site frequency spectrum towards rare variants (Fay et al., 2001; Gao & Keinan, 2014; Booker et al., 2017; and many others). This phenomenon is consistent with our MK test showing that the number of genes under purifying selection is much higher in the domesticated than in the wild populations.

We would like to note here that we did the same test using the same method with the same parameters on the wild and domesticated populations of *S. cerevisiae*, and we found clear difference between the two groups in terms of the intensity of purifying selection and the functional categories of the genes under purifying selection, suggesting that the wild and domesticated populations of the yeast subject to different selection pressure.

As to the effect of “sex” vs “non-sex”, we understand that our finding from the

MK test seems to be inconsistent with the reduced sexuality of the domesticated isolates compared with the wild isolates. It is generally thought that asexual eukaryotes have reduced purifying selection and increased accumulation of deleterious mutations. However, a recent study showed evidence for strong purifying selection in asexual eukaryotes (Brandt et al., 2017, Nat. Commun.). In the case of *S. cerevisiae*, though domesticated isolates have reduced sporulation efficiency and spore viability (implying that they mainly reproduce asexually), we observed more recombination events in the domesticated than in the wild lineages as shown in our structure and LD analyses. As we mentioned in the response to Comment 6 above, Magwene et al. (2011, PNAS) also observed high rates of crossing and mitotic recombination in domesticated isolates. In contrast, the clear population structure and homozygosity of the wild populations implying that wild isolates might mainly undergo homothallic selfing in nature. These observations are probably helpful to explain and understand the significantly increased purifying selection in the domesticated isolates of *S. cerevisiae*. We have discussed this further in the new revised version (Lines 501-510).

As to the effect of mutation rate, we are sorry that we are unable to discuss this reasonably at present because the mutation rates of yeast isolates in nature and fermentation environments are unknown.

10 – GO enrichment on genes under purifying selection (Line:250-257): the link between the GO enrichment results and the interpretation is not clear, and maybe too strongly put. E.g. you are taking a bunch of “different biological processes” that can mean anything, from regulation of protein phosphorylation to cellular polysaccharide metabolic process, from alpha-amino acid metabolic process to cellular amino acid biosynthetic process (just picking from the top 10 of table S7b), suggesting very negative selection at the WHOLE gene regulation level because of nutrient rich environment in domesticated isolates? Line:250-257. Please discuss this further and better or remove.

>> We did the GO enrichment analysis on the genes under purifying selection just for accommodating a comment from Reviewer #1 on the original version: “*It would be interesting to discuss the higher number of genes in purifying selection detected in the domesticated versus wild populations in more detail. How can this signal be linked to differences in population dynamics (expansion vs bottleneck), mating behaviour (sex vs no-sex) and heterozygosity?*” We think it would be helpful to discuss this if we investigate the functions of the genes under purifying selection, but it would be impossible to rely on the exact function of each gene for a set of over 2000 genes. Therefore, we think GO enrichment analysis is probably a feasible way to understand

roughly the biological functions of these genes. We applied the same parameters and class definitions to the wild and domesticated populations. Though the GO classes were defined broadly, we found clear difference between the wild and domesticated populations as shown in Table S7b. For example, enrichment (9.68 fold) of the class meiotic chromosome separation (GO:0051307) was detected in the genes under purifying selection in the wild but not in the domesticated isolates, probably due to the reduced sexuality in the domesticated isolates. We think this result is interesting and can be used to address at least one point of the review comments ‘mating behaviour (sex vs no-sex)’. Thus the discussion related has been remained in the new revised version (Lines 256-259). We think it is more informative to show the GO enrichment of the genes under purifying selection than just to list the names of the genes. Therefore, Table S7b has been remained.

The enrichment of GO classes associated with regulation of different biological processes is much more frequently observed in the genes under purifying selection in the domesticated lineages is also interesting, but the discussion is probably arbitrary and immature, thus we have removed this part in the new revised version.

11- In the previous revision I raised some concerns regarding the Asian origin of all domesticated lineages of *S. cerevisiae*. My text from the first revision: “throughout the manuscript it’s difficult to interpret the major conclusions regarding the origin of domesticated lineages. It would be interesting if the authors would discuss the relationship between the newly identified domesticated lineages in this study and the domesticated lineages identified in previous studies (Liti et al. 2009, Gallone et al., 2016, Goncavales et al., 2016) in more detail. Are the authors implying that all the domesticated lineages originated from a common DOMESTICATED ancestor in Asia that was subsequently spread in Europe and further diversified in the e.g. Wine/Beer lineages? Or rather that WILD *cerevisiae* strains travelled from Asia to Europe, and were domesticated there? Especially the placement of the Milk strains in Figure S1 is an interesting observation in this regard. The first option would go against the hypotheses of Almeida et al. 2015, who identified a wild stock of yeast that they believe are the wild genetic stock of domesticated wine yeasts. The second option also implies that *S. cerevisiae* has been domesticated multiple times in different geographical locations (Europe/Asia), which is maybe more intuitive? Nevertheless, this should be stated more clearly and discussed in more detail”.

>> We revised manuscript accordingly and provided detailed responses to these comments in the last rebuttal letter. We regret that our revisions were not good enough. We have made additional revisions in the new revised version as mentioned in the responses below.

The new version of the text is more complete but still quite confusing. To summarize: based on i) the unusual placement of the Milk clade, ii) partial A,B,C introgression present in CHN-VIII, iii) higher genetic diversity in the milk clade, the authors conclude indeed that all the domesticated lineages documented so far originate from Far East Asia and that the Wine yeasts were transferred from Asia (and also Beer1 and MO sister lineages of the Wine lineage?). Do the authors imply that the MO clade is not the wild stock of the Wine clade?

>> These specific points are additional evidence supporting or being consistent with our hypothesis. We propose the Far East Asian origin hypothesis mainly based on our phylogenomic analyses on the whole wild and domesticated isolates of *S. cerevisiae* sequenced in this and other studies, showing that Far East Asia is the center of origin for both the wild and domesticated populations of the yeast species. Please see our detailed responses below.

One point I do not agree with is the statement that higher nucleotide diversity indicates necessarily an earlier origin. The authors use this argument e.g. to prove that the milk lineage is older than the wine lineage: 'The genetic diversity of the Milk lineage ($\pi = 3.86e-03$, 15 isolates) is much higher than (more than three times) that of the Wine/European lineage ($\pi = 1.04e-03$, 9 strains in Liti et al. 2009; $\pi = 1.12e-03$, 19 isolates in Almeida et al. 2015; and $\pi = 1.59e-03$, 24 isolates in Gallone et al. 2016) as shown in Figures 1 and S1 and Table S3. The data suggest that the Milk lineage is native in Asia.' Or 'First, if the MO lineage is the wild genetic stock of the Wine lineage, the nucleotide diversity of former should be significantly higher than that of the latter. However, Almeida et al. (2015) showed that the genetic diversity of the MO lineage ($\pi = 0.99E-03$, $n = 31$) is lower than that of the Wine lineage ($\pi = 1.12E-03$, $n = 19$).'

Everything depends on the life history of the yeasts. More generations (e.g. because of serial re-inoculation of the yeasts, such as in beer or milk lineages), automatically results in faster evolution, and thus diversity, compared to yeasts that only undergo a few generations per year (such as the wine yeasts).

>> The genetic diversity comparison is only one point of the evidence, which is consistent with our hypothesis. We agree that the life history has effect on the genetic diversity of yeast, especially the domesticated isolates associated with different fermentation processes. But we think that it is indispensable to compare genetic diversity for a population genomic study.

In the case of the MO vs. the Wine lineages, even though the genetic diversity of wine isolates is probably compromised by their limited generations in a year, they still

have higher genetic diversity than the MO lineage, being not consistent with the hypothesis that the Wine lineage originated from the MO lineage.

We think the clustering of the MO lineage within the domesticated liquid-state fermentation group is interesting but in our opinion the origin of the lineage remains to be addressed further. Several isolates sampled from Mediterranean oaks were located in the Wine lineage (Almeida et al., 2015). Interestingly, we found that one MO isolate EXF7145 (see Figure S3 in Almeida et al., 2015) is clustered in the Milk lineage in our analysis (please see the tree below). Isolate EXF7145 also contains exactly the same introgressed *GAL7-GAL10-GAL1* cluster as the Milk isolates, suggesting that this MO isolate probably came from the population associated with milk fermentation. These data suggest that Mediterranean oaks can harbor isolates from diversified sources including fermentation processes. We think that it is worthy of further studying to reveal the origin of the Mediterranean oak population of *S. cerevisiae*.

ML tree constructed from 149,092 SNPs, showing the position of a Mediterranean oak isolate EXF7145.

In the case of the Milk vs. the Wine lineages, we added the comparison to accommodate the review comments from Reviewer #1 on the first version: *Figure S1 shows that these (the Milk) strains are in a sister lineage of the Wine strains, and thus must have diverged from the Wine clade AFTER they split from eg. the Beer 1 clade. Therefore, it is intuitively more likely that these Milk strains are more commercial/industrial strains that were initially domesticated in Europe and later transferred back to Asia, no? Or how do the authors interpret this phylogeny?* In addition to provide the information about the source of the Milk isolates which are all from spontaneously fermented dairy products sampled from local families in different remote regions of China and Mongolia, we think it is necessary to compare the genetic diversity of the two lineages. We believe that the much higher genetic diversity of the Milk lineage than that of the Wine lineage supports our hypothesis that the Milk isolates are traditionally domesticated isolates native to Asia, rather than ‘commercial/industrial strains that were initially domesticated in Europe and later transferred back to Asia’. The phylogeny shows that the Wine lineage share a common origin with the Milk lineage, we therefore infer that the former should also originate from Asia. This hypothesis is supported also by other findings including the clustering of Chinese wine isolates in the Wine lineage and the sharing of rare HGT events of European wine isolates with Chinese wine and wild isolates.

Also regarding the HGT. The authors state: “Third, European Wine isolates share HGT genes with Chinese isolates. Among the three horizontally transmitted regions (A, B and C) which are regarded as domestication fingerprints of wine isolates, one (region B) is present in the wild lineage CHN-VIII and a partial fragment of region C is present in three Chinese isolates in the Wine lineage (Figure 4, Table S8). These HGT genes are absent from the MO lineage (Almeida et al., 2015).”. “The close relationship of the Mediterranean oak (MO) group with the Wine lineage was also resolved (Fig. S2 and supplementary information). The two sub-groups diverged directly from a common ancestor (Fig. S1, S2).”

It’s not really clear to me how this shows an Asian origin. Looking at the close relationship between the Wine, MO, Beer2 clade and Milk, are the authors implying that these HGT regions were lost in the sister lineages of the Wine clade? It’s hard to spot the actual relationships between strains from the radial phylogeny in Fig. S2. The point is that I probably believe in the Asian origin of the domesticated lineages. But the authors fail to explain this hypothesis clearly in the text; they base their assumptions on hints that are very interesting per se’ but are put together in a confusing way without considering all possible implications.

>> Again, we do not propose the Asian origin hypothesis based only on the HGT

events, we use them to support this hypothesis and to show that the HGT events are consistent with our hypothesis. We believe that the sharing of the rare HGT events of European wine isolates with Chinese wine isolates and Chinese wild isolates in the CHN-VIII lineage (the closest wild relative of the domesticated lineages) supports or is consistent with the Asian origin hypothesis of the Wine lineage. Given the origin of the domesticated lineages from the wild and the gene flow generally from the wild to the domesticated populations, it is unlikely that the Chinese wild isolates obtained the HGT fragment for European wine isolates (Lines 394-396). The absence of these HGT events in the Beer 2, Milk and MO clades is another problem which does not conflict with the Asian origin hypothesis of the Wine lineage.

Finally, we appreciate that Reviewer #1 actually believes in the Asian origin of the domesticated lineages and we thank his/her detailed comments and suggestions for improving the quality of our manuscript. For a better understanding, here we would like to summarize briefly our arguments for the Far East Asian origin hypothesis of the domesticated lineages of *S. cerevisiae*. The two main reasons are:

- 1) We showed evidence for a Far East Asia origin of the species *S. cerevisiae* (this region harbors the highest genetic diversity and the oldest lineages of the species documented so far in the world); and
- 2) We showed that Far East Asia is also the center of origin of the domesticated populations of *S. cerevisiae* (much higher genetic diversity and much more distinct lineages in this region than in other regions of the world investigated). Among the 16 distinct domesticated lineages recognized so far worldwide (Wine, Beer 1, Beer 2, ADY, Baijiu, Huangjiu/Sake, Qingkejiu, Milk, Mantou 1 to Mantou 7 and the Mixed lineage), 12 lineages were found only in Far East Asia.

We used more evidence to show or to infer that the other four domesticated lineages (Wine, Beer 1, Beer 2, and the Mixed) also likely originated from Asia.

- 1) The domesticated lineages form two monophyletic major groups: the liquid- and the solid-state fermentation groups. The former containing the four foreign lineages share a recent ancestor with the latter containing only Asian lineages.
- 2) Within the liquid-state fermentation lineage, the Wine lineage originated from the same ancestor with the Milk lineage which is native to Asia.
- 3) The Wine lineage also contains Chinese isolates which share rare HGT events with European wine isolates. The Wine lineage also shares rare HGT events with Chinese wild isolates. China has longer winemaking history as showed by archaeological evidence.

4) For the Beer lineages, though we have not sampled beer yeast from China, archaeological evidence showed that beer production history in China is longer than in Europe.

The points mentioned above are all showed and discussed in our manuscript. We hope the reviewer will find our evidence and discussion are strong and acceptable for the Far East Asian origin hypothesis of the domesticated populations of *S. cerevisiae*.

Reviewer #2 (Remarks to the Author):

I was satisfied with this revision.

>> We thank Reviewer #2 for his all comments and time.

Reviewer #3 (Remarks to the Author):

The manuscript is greatly improved and will be a significant advance in our understand of *S. cerevisiae* diversity. There are, however, some important concerns with the language/interpretation of the results. In some cases I believe a slight change in the wording will provide the necessary accuracy. In other cases it may be a change in the interpretation. It was always clear to me whether this was language or interpretation.

>> Thanks for this generally positive comments.

The two main issues are: the 'single or two ancestors' statements and purifying selection in the domesticated group, highlighted by (*) below.

1) I suggest these changes to the abstract for better grammer/clarity:

A previous study showed that wild populationS of *S. cerevisiae* probably originated in Far East Asia. However, the diversity and evolutionary history of domestic populationS of the yeast remain elusive.

>> Done.

we show here that China/Far East Asia is also likely the CENTER OF ORIGIN of domestic populations of the species

>> Done.

The domestic populationS...

>>Done.

2) “The fraction of the ancestral population that entered into the wild and the domestic group is 99.36 % and 0.64 %, respectively.”

This isn't clear. Is this the amount of variation that survived the bottleneck. What parameters are specifically estimated? The bottleneck during, change in N?

>> The demographic history was analysis based on genome wide non-coding SNPs using software package $\partial a \partial i$. The ancestral population is represented by the estimated effective population size ' N_A '. The data mean that only $0.64\% * N_A$ entered into domesticated group while the rest ($99.36\% * N_A$) remained in the wild population. We have added “estimated effective size of the ancestral population” in the revised version (Lines 172-173) and the SI part (Lines S218-S219).

3) “with an average ratio of heterozygous sites of 0.0055%”

This statement isn't clear. Only in the methods is it mentioned it is the ratio to the genome size. Better to call it the proportion of heterozygous sites in the genome.

>> We have added “in the genome” (Line 183).

4) should be: “, consistent with Magwene et al. (2011)”

>>Done.

5) should be: “only specific lineages; most remarkably,”

>>Done.

6*) The language used to describe MK test is not accurate. The MK test does not determine whether a gene is subject to selection. Indeed, a non-significant MK test is consistent with purifying (negative) selection, which is consistent with a neutral model. The MK tests whether observed polymorphism/divergence is subject to selection. The standard neutral model outlined by Kimura is not rejected by MK test indicating $NI > 0$. $NI < 0$ is not consistent with neutral model.

>> We are not sure the meaning of $NI > 0$ or $NI < 0$, because $NI = (P_n/P_s)/(D_n/D_s)$ which will always > 0 and will never < 0 . Probably the viewer actually wanted to say the fraction of nonsynonymous substitutions driven by positive selection ($\alpha = 1 - NI$). The null hypothesis of the McDonald-Kreitman test is neutrality (McDonald and Kreitman 1991; Smith & Eyre-Walker, 2002; Fay et al., 2002; Egea et al., 2008; Fay, 2011; and many others). Under neutrality, P_n/P_s equals D_n/D_s and thus $NI = 1$ ($\alpha = 0$). If supported statistically, $NI > 1$ (or $\alpha < 0$) indicates an excess of amino acid

polymorphism due to negative or purifying selection; while $NI < 1$ (or $\alpha > 0$) indicates an excess of nonsilent divergence due to positive selection.

Prior interpretations of the MK test have been relaxed purifying selection in domesticated lineages leading to more amino acid polymorphism than expected. The way the manuscript is written, it sounds like there is more purifying selection in domestic lineages.

>> Our data suggest more purifying selection in the domesticate lineages. We found that more than half (**51.5%**) of the genes subjected to the MK test demonstrated purifying selection in the domesticated population. In contrast, only **8.9%** of the genes tested demonstrated purifying selection in the wild population (Tables S3 and S7). Please see our comments on the prior study below.

should be: “the remaining (91.0%) were consistent with a neutral model”

>> OK, we have revised this sentence accordingly (Line 255).

The neutral model includes purifying selection.

>> Anyway, purifying (negative) selection and positive selection are the two main types of natural selection leading to adaptive evolution.

And in the discussion “purifying selection are rare in the wild population,” This is not supported. Purifying selection is stronger in wild than domestic populations as established by prior work and supported by the authors own analysis. See comments on MK test.

>> Here we mean that purifying selection is rare in the wild population as compared with the domesticated population. We say this based on our data obtained from the MK test. As mentioned above, our data show that the ratio of genes subject to purifying selection in the wild population (**8.9%**) is much lower than that in the domesticated population (**51.5%**), suggesting that purifying selection is stronger in domesticated than in wild populations. Comparison of minor allele frequency distributions also supports a stronger purifying selection in the domesticated population (please see the response to Comment 9 of Reviewer #1 above).

As we mentioned in the response to Comment 5 of Reviewer #3 on the original version concerning the MK test, the previous study (Elyashiv et al., 2010) compared only 16 *S. cerevisiae* strains, including 8 European, 3 Sake, 3 Malaysian, 1 North American and 1 West African strains, and really wild population was not included. The polymorphisms within each lineage compared in the prior work were very limited. Since the MK test and other population genetic analyses heavily rely on statistical

analysis based on large samples, we think that the results obtained from very limited samples with very limited polymorphisms should be interpreted with caution. In our study we compared 106 wild and 160 domesticated isolates and thus we believe our result is more reliable.

7) should be: “C) first found in the wine yeast strain”

>> Done.

8*) Are the population genetic analyses inaccurate due to introgression? I worry that some SNPs calls are for introgressed regions, e.g. from Spar into strain x, yielded artificially high divergence and rates of polymorphism. In particular, CHN-IX, which shows the most extensive introgression with *S. paradoxus*.

>> This possibility was excluded in our population genetic analyses. The SNPs were obtained from variant calling based on reference-based alignment. The clean paired reads were mapped to the reference S288c (R64-1-1) genome using the Bowtie2 program with default settings, which excluded the reads with < 97.3% nucleotide identity with the reference genome. The introgressed regions usually exhibited < 95% sequence identity with the reference genome and thus were not included in the SNP calling.

9*) “and each group originates directly from a single recent ancestor (Figs. 1, S1 and S2), implying that the domestic lineages recognized worldwide so far can be traced back to no more than two ancestors.”

I'm sorry, but this statement is not supported. When there is no recombination one can trace lineages back to a single common ancestor, e.g. human mtDNA, but this does not imply a bottleneck or anything related to domestication; it is a simple consequence of population genetics and entirely expected. When there is recombination there are multiple ancestors.

I'd suggest changing the wording to .. originating from a single lineage or single population.

That all domesticated yeasts come from either of two lineages/populations is a significant statement – even if very much consistent with most but not all prior work. But to come from one or two ancestors (cells implied) is incorrect or not supported.

“lineages recognized so far originate from a single isolate that” I'd say they have a shared origin (population or lineage implied or explicitly stated), not a single isolate which is unlikely and no data to support or refute.

Similar statements elsewhere would need to be changed.

A general – 'single origin hypothesis' is fine. The supporting MAL and FLO gene analysis provide compelling evidence for a shared origin even if subsequent admixture occurred.

>> We say “each group originates directly from a single recent ancestor” based on the results of our phylogenomic analyses (Figs. 1a, S1 and S2) clearly showing that each of the liquid- and solid-state fermentation groups forms a monophyletic group. **A monophyletic group is defined as a group of organisms that forms a clade which consists of all the descendants of a common ancestor.** We have modified the sentence to “each group originates directly from a recent common ancestor” (Line 358).

Furthermore, we show that the liquid- and solid-state fermentation groups containing all the domesticated lineages form a major monophyletic group which clearly separated from the wild lineages, suggesting that all the domesticated lineages recognized so far also share a common ancestor. In addition, we provide more evidence including the sharing of CNV patterns of MAL, FLO and other genes of the domesticated lineages. Therefore, we propose the single origin hypothesis.

We appreciate that the 'single origin hypothesis' is accepted by the reviewer. We thank the suggestions of the reviewer for improving the wording. We agree that it is not appropriate to say “the domestic lineages recognized so far originate from a single **isolate**”. We have deleted this sentence in the new revised version.

We say in the manuscript “that the two major domesticated groups share a common ancestor that diverged from the wild lineage CHN-VI/VII ...”. In agreement with Magwene et al. (2011) and Magwene (2014), our results suggest that the ancestor of the domesticated lineages was likely formed by outcrossing between genetically different wild isolates, because of the sharing of heterozygosity by all the domesticated lineages and the sharing of homozygosity by all the wild lineages. Thus, our hypothesis encompasses recombination.

The possibility that the crossed wild isolates forming the heterozygous ancestor came from different wild lineages (though most likely closely related) cannot be excluded. Therefore, we think it is probably inaccurate to say that the domesticated lineages originated from a single lineage or single population. Such an ancient lineage or population consisted of heterozygous isolates has not been identified. Nonetheless, we think ‘an ancestor’ can be understood as ‘an ancient lineage/population’.

Another consideration on the use of the word ‘ancestor’ is the comparison with the

result of a recent study. Gallone et al. (2016, Cell) concluded that “today’s industrial yeasts originate from only a few ancestors”. Our result suggests that the domesticated yeast lineages originate from no more than two, most likely one recent ancestor (the single origin hypothesis).

10) typo: “all most all the domestic”

>> Done.

11) should be: “diversity of the former should”

>> Done.

12) Regarding the MO lineage, there are some caveats to the statements made. The higher diversity of wine lineages could be subsequent admixture where MO has had no admixture. The wine-like strains found in china could be migrants from europe. In my view, the last argument is strong, but should be flushed out. The HTG genes in china that define the wine group either formed in europe and then had to have migrated to china, which seems quite unlikely. Or the original HTG occurred in china and the wine strains were derived from this domestic population.

>> Thanks for pointing out the caveats. For the genetic diversity of the Wine lineage, Reviewer #1 points out an opposite possibility that wine yeasts only undergo a few generations per year and thus produce less diversity. Nevertheless, we have deleted the comparison of genetic diversity between the Wine and MO lineages, though we think the data support our hypothesis.

The European wine isolates also share a HGT event with the Chinese wild lineage CHN-VIII. It is unlikely that the Chinese wild isolates obtained the HGT fragment for European wine isolates, given the origin of the domesticated lineages from the wild and the general gene flow from the wild to the domesticated populations. This sentence has been added to the new revised version (Lines 394-396).

Please see our response to the last comment from Reviewer #1 for our arguments on the origin of the Wine as well as other domesticated lineages.

13) Typo “while a considerable NUMBER? of other genes”

>> Corrected.

14) should be “may have AN advantage”

>> Corrected.

REVIEWERS' COMMENTS:

Reviewer #1 (Remarks to the Author):

The authors have addressed all our remaining concerns and suggestions and we therefore wholeheartedly support publication of this interesting paper. In fact, we wish to thank the authors for their efforts to respond to our suggestions; we actually believe that this really made a good study even better.

best regards,
Brigida Gallone, Jan Steensels and Kevin J. Verstrepen

Reviewer #3 (Remarks to the Author):

In the previous review I advocated for toning down the rather extreme interpretations of domesticated strains coming from a one/two ancestors. I still do not believe this interpretation is supported and using the term monophyletic is worse than the language used in the prior version. At best the populations do have one or two ancestors, at worst the assumptions of the tree (no recombination) are violated and the group may only appear to be monophyletic but carry bits and pieces of ancestry from wild populations. I remain skeptical that there has not been gene flow with wild populations. Indeed, the authors describe gene flow from wild to domesticated populations: pg 14, line 396. If there is recombination and gene flow with wild populations, they are not monophyletic and do not have one/two ancestors.

Regarding selection, I still maintain that there is not more negative selection in the domesticated population. There are more deleterious mutations, indicated by $NI > 1$. This can be interpreted by a simple bottleneck whereby weakly deleterious mutations can drift to high frequency during the bottleneck. Changes in population size are known to affect NI: see PMC1462352.

Sorry for the typo in previous review, the authors are correct that NI comments should be about $< > 1$ not $< > 0$.

Responses to Reviewers' comments:

Reviewer #1

The authors have addressed all our remaining concerns and suggestions and we therefore whole heartedly support publication of this interesting paper. In fact, we wish to thank the authors for their efforts to respond to our suggestions; we actually believe that this really made a good study even better.

>> We heartedly thank Reviewer #1 for his/her time and all invaluable comments, which have helped us to improve the quality of the manuscript substantially.

Reviewer #3 (Remarks to the Author):

1 –In the previous review I advocated for toning down the rather extreme interpretations of domesticated strains coming from a one/two ancestors. I still do not believe this interpretation is supported and using the term monophyletic is worse than the language used in the prior version. At best the populations do have one or two ancestors, at worst the assumptions of the tree (no recombination) are violated and the group may only appear to be monophyletic but carry bits and pieces of ancestry from wild populations. I remain skeptical that there has not been gene flow with wild populations. Indeed, the authors describe gene flow from wild to domesticated populations: pg 14, line 396. If there is recombination and gene flow with wild populations, they are not monophyletic and do not have one/two ancestors.

>> We are sorry that we still respectfully disagree with Reviewer #3's opinion "*If there is recombination and gene flow with wild populations, they (domesticated lineages) are not monophyletic and do not have one/two ancestors*". If our understanding is correct, Reviewer #3's opinion is that because domesticated lineages likely have genes from different wild isolates because of recombination and gene flow, it is not correct to say that the domesticated lineages originate from a single ancestor. According to this opinion, none of the *S. cerevisiae* lineages showed in our study, no matter how large or how small, can be said to have a common ancestor, because they mostly have more or less alien genes from different species as we show in Figure 4.

Let's use an example to show the problem of this opinion. Laboratory *S. cerevisiae* strains are mostly derivatives of strain S288C. They have been subjected to different genetic modifications by recombination and other protocols in different laboratories all over the world (similar to domesticated isolates subjected to different human selection for different fermentation processes) and thus must carry various genes from different strains or species. However, if we sequence all the derivative strains and do a phylogenetic analysis based on the whole genome sequences and use wild isolates,

say oak isolates, as the outgroup, all the derivatives of S288C must form a monophyletic clade. We think it should be correct to say that these laboratory strains originate from a single ancestor S288C, though they carry genes from different sources. This is the logic of our interpretation based on our phylogenetic analyses using the genome wide SNPs. Our data clearly show that all the domesticated lineages form a major monophyletic clade, which contains two monophyletic sub-clades (the solid- and liquid-state fermentation groups). They do carry many fragments or genes from many different species within and outside the genus *Saccharomyces* as we show in our manuscript, but we can not say the latter are all ancestors of the domesticated lineages of *S. cerevisiae*.

Nevertheless, we respect the comments from Reviewer #3 and have toned down the one/two ancestor origin hypothesis of the domesticated populations. We have deleted 'one/two ancestors' from the abstract. Considering Reviewer #3 accepts the single origin hypothesis of the domesticated lineages as he/she wrote "*A general – 'single origin hypothesis' is fine*" in his/her comments on the last version, in the main text, we have changed 'a common ancestor' to 'a common **origin**' in the third revised version when referring to the origin of the whole domesticated population.

2 –Regarding selection, I still maintain that there is not more negative selection in the domesticated population. There are more deleterious mutations, indicated by $NI > 1$. This can be interpreted by a simple bottleneck whereby weakly deleterious mutations can drift to high frequency during the bottleneck. Changes in population size are known to affect NI: see PMC1462352.

>> Sorry we do not agree that "*there is not more negative selection in the domesticated population*" because our data show the opposite. Considering the opinion of Reviewer #3, we have removed the result of the MK test from the main body. We have presented strong evidence showing that ecology is the primary force driving the diversification of the domesticated population of *S. cerevisiae* (adaptive evolution driven by natural selection). Removing this part will not influence our main conclusion.

However, we think that the data are interesting, which were also acknowledged by Reviewer #1 who wrote "The MK test: the much higher number of genes under purifying selection is per se' very interesting" in his/her last comments. We thus did not completely deleted the MK test result but removed this part and related discussion to the Supplementary Information as Supplementary Note 5 and remained the table showing the MK test result as Supplementary Data 9.